# The impact of blockchain adoption on supply chain financing and E-commerce platform dynamics

**Shangyu Pi**[iD]*

School of Management, Guangdong Polytechnic Normal University, Guangzhou, China

* shangyupi510665@163.com

**Citation:** Pi S (2026) The impact of blockchain adoption on supply chain financing and E-commerce platform dynamics. PLoS One 21(1): e0339597. https://doi.org/10.1371/journal.pone.0339597

## Abstract

The integration of blockchain technology is reshaping e-commerce supply chains and creating new dynamics for financially constrained firms. This study investigates the operational and financial implications of blockchain adoption within a supply chain consisting of a financially constrained manufacturer and a dominant e-commerce platform. Our analysis spans three common entry modes—self-operated, Fulfillment by Platform (FBP), and Sales on Platform (SOP)—and considers two key financing schemes: bank financing and equity financing. Using a game-theoretic framework, we find that while blockchain generally enhances profitability, its strategic value is highly contingent upon the chosen entry mode. Specifically, the self-operated mode grants the platform greater flexibility in pricing and service, whereas the FBP and SOP modes lead to more stable operational decisions. Our analysis further reveals a critical interplay between financing and operations: platforms maximize their gains by pairing the self-operated mode with equity financing, while a manufacturer's optimal financing strategy shifts from bank to equity as production costs rise. These findings offer actionable insights for managers on leveraging blockchain to optimize financing choices and enhance competitive advantage. This study is, however, limited by certain simplifying assumptions, including linear demand and a uniform consumer response to blockchain, and its scope is restricted to bank and equity financing.

## 1 Introduction

The digital economy has firmly established e-commerce platforms as a principal sales channel for manufacturers. This is not a marginal shift but a profound global trend, evidenced by explosive market growth. In 2023 alone, U.S. online retail sales surged to $1.295 trillion, while China's market expanded to 15.4 trillion yuan. This expansion is mirrored by industry leaders. Amazon's net sales exceeded $574 billion, and major brands like Nike now generate over a quarter of their total revenue from digital channels, a significant leap from just 10% in 2019. This rapid ascent is driven by two core factors: the inherent power of platforms and their flexible sales

**Data availability statement:** All relevant data are within the paper and its Supporting Information files.

**Funding:** 1. Institutions Young Innovative Talent Program (2023WQNCX039), Sponsor: Department of Education of Guangdong Province; 2. 2021 Guangdong Polytechnic Normal University Talents Research Project, Sponsor: Guangdong Polytechnic Normal University. The funders had no role in study design, data collection and analysis, decision to publish, or preparation of the manuscript.

**Competing interests:** The authors have declared that no competing interests exist.

formats [1]. Platforms transcend the physical constraints of brick-and-mortar retail, creating powerful network effects that attract a growing base of users and manufacturers [2,3]. Furthermore, they enhance profitability through diverse operating models, most commonly the wholesale model (where the platform is a reseller) and the agency model (where the platform is a marketplace) [4,5]. While the agency model can stimulate demand through lower prices, the wholesale model grants platforms greater pricing power, albeit with the risk of double marginalization [6,7]. Despite this online momentum, traditional retail retains key advantages, meaning the synergistic development of these dual channels remains a critical area of study [8].

As consumers' living standards improve, their focus is shifting from price alone to a broader consideration of sales and service quality. Consequently, the price advantage of a given channel is diminishing, while the importance of service has become a central concern for both consumers and supply chain members. Sales service, as a value-added component of products, is now regarded as a crucial marketing tool for enhancing consumers' willingness to pay [9]. Such services can be categorized based on the sales stage. Pre-sales services, for instance, inform consumers about products by providing consultations [10,11]. In-sale services (e.g., payment guidance, packaging, and logistics) and after-sales services (e.g., installation, commissioning, and technical guidance) can increase consumer satisfaction and strengthen brand competitiveness [6,12]. In a platform-based supply chain, sales services may be provided by either the brand manufacturers or the platforms themselves [13]. Intuition and online purchasing experiences suggest that different service delivery strategies vary in efficiency, leading to different consumer experiences [1]. Consumers with high service preferences may even be willing to pay higher prices for superior service quality [14,15]. However, this ecosystem is often plagued by information asymmetry and a lack of transparency, leading to risks like false advertising and poor service. This underscores the critical importance of accurate, traceable information for all supply chain participants.

Blockchain technology offers a powerful solution to these trust-related challenges. As a decentralized and transparent database, its core advantage is ensuring mutual trust while reducing the costs associated with it [16]. This potential is already being realized on major e-commerce platforms. For instance, Walmart uses blockchain to ensure the traceability of fresh goods, while JD.com has deployed its own blockchain to guarantee the authenticity of high-value products like jewelry [1,17]. Furthermore, IBM has collaborated with Veridium Labs to use blockchain technology for tracking carbon footprints [18]. These applications demonstrate blockchain's capacity to solve critical information-related problems in the supply chain.

However, another pressing challenge exists within the platform ecosystem: the financial constraints of small and medium-sized enterprises (SMEs). SMEs are vital suppliers for e-commerce platforms, yet they often struggle to secure loans from traditional banks due to a lack of collateral and credit history [19–21]. To maintain a stable supply chain, platforms like Amazon and JD.com have stepped in to offer financing services directly to their merchants [22]. Consequently, bank credit and platform-based financing have emerged as the two dominant financing avenues for these capital-constrained manufacturers.

Based on the preceding observations, this study investigates the integration of blockchain technology into the operational and financing strategies of platform-based supply chains. We focus on a manufacturer facing financial constraints and analyze their service and financing decisions across three prevalent entry modes, exemplified by the JD.com platform. The first is the self-operated mode, which functions as a wholesale model. The other two modes are subdivisions of the agency model: the Fulfillment by Platform (FBP) mode, where the platform provides key services such as logistics and after-sales support, and the Sales on Platform (SOP) mode, where the enlisted merchant manages these services independently. To this end, we construct a game-theoretic model to analyze the supply chain's financing, service, and pricing decisions under different blockchain adoption scenarios. The primary objectives of this analysis are to determine the impact of blockchain technology on supply chain operations across the three distinct entry modes and to identify the conditions under which adopting blockchain is strategically beneficial for the supply chain participants.

This paper aims to answer the following research questions:

1. How does the adoption of blockchain technology influence service decisions within the supply chain?
2. How does blockchain technology affect manufacturers' choices of financing strategies?
3. What is the impact of blockchain technology on the profitability of supply chain members within a dual-channel sales framework?

By addressing these questions, this study provides insights into the strategic implications of integrating blockchain technology for service management, financing decisions, and profitability in the context of dual-channel sales.

The primary contributions of this paper are summarized as follows:

1. This paper contributes to the literature by integrating blockchain technology with an analysis of different platform entry modes. We explicitly model how blockchain adoption influences consumer service preferences and, consequently, shapes supply chain service decision-making. This approach offers a comprehensive framework for evaluating the operational impact of blockchain, providing valuable insights for supply chain management in the digital economy.
2. Methodologically, this paper advances the literature by moving beyond the predominantly qualitative studies on blockchain. We construct a quantitative, game-theoretic model to analyze the technology's impact on dual-channel supply chain operations. A key innovation is our explicit modeling of how blockchain influences financing costs under different schemes, providing a rigorous and operable framework for quantitatively assessing the integration of blockchain into supply chain management.
3. This study provides significant managerial and policy implications. We conduct a detailed comparative analysis of key operational and financial outcomes (including services, pricing, financing choices, and profits) under scenarios with and without blockchain. Crucially, our analysis identifies the specific conditions that make blockchain adoption strategically advantageous. These findings offer a theoretical foundation to guide practitioners' decision-making and provide valuable insights for policymakers aiming to promote the development of supply chain finance and establish supportive regulatory environments.

## 2 Literature review

This study is situated at the intersection of three primary research streams: e-commerce platform entry modes, service decisions in dual-channel supply chains, and the application of blockchain technology to supply chain operations. In the following sections, we will critically review the existing literature within each of these domains to build a logical foundation for our research and highlight its unique contributions.

## 2.1 E-commerce platform entry mode

Entering e-commerce platforms has become a cornerstone strategy for supply chain enterprises seeking to navigate the new retail landscape and unlock new profit growth [23]. When partnering with these platforms, manufacturers face a critical choice between two primary sales models: the wholesale model and the agency model [24,25]. In the wholesale model, the platform acts as a traditional retailer, purchasing products in bulk and reselling them to consumers, as exemplified by JD.com's self-operated business. In contrast, the agency model allows manufacturers to operate their own storefronts, selling directly to consumers while the platform charges a commission—a structure famously used by brands like Nike and Under Armour on Tmall.

This choice of sales model directly dictates the allocation of service responsibilities. While the wholesale model typically sees the platform managing most sales services, the agency model presents a more complex decision. Here, service responsibility can be either retained by the manufacturer (SOP mode) or delegated to the platform (FBP mode) for functions like logistics and after-sales support [26,27].

Research on this strategic choice, while nascent, has begun to explore the trade-offs between service control and profitability. From a service perspective, Wang and Yan [26] argued that in complex e-commerce environments, optimal outcomes are often achieved when merchants, rather than the platform, provide key services, suggesting a preference for the SOP mode under certain conditions. However, from a profitability standpoint, the picture is more nuanced. A comparative study by Wang et al. [27] analyzed the FBP and SOP modes and revealed a potential conflict of interest: platforms generally earn higher profits under FBP, while suppliers' preference for FBP emerges only when commission rates are sufficiently high. Taken together, while these studies provide valuable insights into the platform-supplier dyad, they predominantly analyze these entry modes in isolation.

Despite these valuable contributions, the literature has yet to fully explore how the choice of entry mode interacts with a manufacturer's own offline channel, particularly concerning the strategic delegation of service responsibilities. This gap highlights the importance of examining service decisions within dual-channel supply chains, which we address next.

## 2.2 Service decision-making in dual-channel supply chain

The literature on dual-channel supply chains has increasingly recognized the pivotal role of e-commerce platform services in shaping competitive strategies. An initial stream of research confirms that platform services are a powerful lever for value creation. For example, Modak et al. [28] demonstrated that centralized decision-making, which better coordinates these services, yields higher overall returns. This finding is reinforced by Yi et al. [29] and Wang et al. [30], who showed that platform-provided value-added services can enhance both profitability and customer value, although consumer price sensitivity can complicate pricing strategies. More recently, Zhao and Wang [31] introduced behavioral factors, revealing that a retailer's equity concerns can further influence service levels.

While insightful, this body of work predominantly focuses on traditional hybrid channels (offline-online), leaving a critical context underexplored: competition between two distinct online channels. When both sales channels operate digitally, the dynamics of service coordination and communication become uniquely critical. The literature addressing this specific context is sparse. Among the few relevant studies, Zhang et al. [32] and a subsequent study by the same authors [33] began to map this territory, comparing cooperative service scenarios and the impact of new tools like live-streaming.

Despite these advances, it remains underexplored how the choice of a supplier's entry mode interacts with the platform's service strategy to shape overall service decisions within a dual-channel framework. This gap in the literature motivates a closer examination of the interplay between entry mode selection and service coordination, particularly in the presence of enabling technologies like blockchain.

 

## 2.3 Application of blockchain technology in supply

Blockchain technology is emerging as a transformative force in supply chain management, primarily valued for enhancing traceability, transparency, and transaction efficiency. Early research has established its benefits in niche but important areas. In green supply chains, for instance, its ability to track carbon footprints and enable trusted product information has been shown to boost consumer trust and pro-environmental behavior [34]. Similarly, in supply chain finance, "blockchain + finance" models are celebrated for accelerating financing processes by improving transparency and reducing transaction costs [35,36].

Empirical studies have begun to validate the positive market effects of this enhanced transparency. Wu et al. [37] and Liu et al. [38] confirmed that blockchain-enabled information disclosure can directly drive consumer purchasing behavior for sustainable products. Nevertheless, the path to adoption is not without friction. High investment costs and deployment barriers remain significant hurdles, as noted by Babaei et al. [39]. Despite these challenges, the value proposition remains strong in specific contexts, with studies like Bhatia et al. [40] and Sharma et al. [41] continuing to pinpoint its role in reducing transaction costs and improving performance in sectors like agriculture.

Despite this growing body of research, two significant gaps remain, which this study aims to address. First, while studies have explored blockchain's role in financing and coordination, particularly in the context of consumer low-carbon preferences, they have largely overlooked how these dynamics are affected by a manufacturer's choice of e-commerce platform entry mode. Second, the literature on dual-channel supply chains, even when addressing sustainability issues like emission reduction, has yet to systematically investigate the impact of blockchain technology on core operational decisions, including sales and service strategies. This paper bridges these gaps by developing a model that integrates blockchain adoption with different platform entry modes and their associated service and financing decisions.

To conclude this review, Table 1 positions our study's key contributions relative to the existing literature. In addressing the identified research gaps, our model uniquely integrates a manufacturer's choice of e-commerce platform entry mode with considerations for consumer preferences toward both low-carbon attributes and service quality, all within a dual-channel sales framework. The resulting analysis aims to provide a theoretical foundation that can guide low-carbon supply chain enterprises in their strategic decisions regarding emission reduction, service levels, and financing.

The remainder of this paper is structured as follows: we first present the model construction, followed by numerical simulations and case studies. We then discuss the managerial insights derived from our findings and, finally, conclude with a summary of the study and directions for future research.

## 3 The model

We consider a two-level supply chain consisting of a single financially constrained manufacturer (M) and a dominant e-commerce platform (P). The strategic interactions between these two agents are modeled as a Stackelberg game, allowing us to analyze their sequential decision-making processes.

Our analysis is conducted across three distinct platform entry modes, which are representative of common industry practices: a self-operated mode, a FBP mode, and a SOP mode. Within each of these settings, we investigate the manufacturer's optimal production and service level decisions. A central element of our analysis is the manufacturer's choice between two financing options: traditional bank financing and platform-based equity financing.

Furthermore, we explicitly model the strategic decision of whether to adopt blockchain technology. To enhance clarity, the sequential decision-making process for each model is visually summarized in the conceptual flow diagram in Fig 1. This allows us to compare the equilibrium outcomes—including production, service, and financing choices—and the profitability of the supply chain members both with and without its implementation. Our model also incorporates key consumer behavioral factors, including their channel preferences and sensitivity to service levels. The key parameters and notations used throughout the analysis are defined in Table 2.

Table 1. A brief literature survey.

| Author(s) | Blockchain Technology Adoption | Market entry Approaches | Service Level | Financing Choices | Dual-Channel Supply Chain |
|---|---|---|---|---|---|
| Modak et al. [28] | | | √ | | |
| Xiao et al. [23] | | √ | | | √ |
| Wang et al. [30] | | | √ | | √ |
| Wen et al. [24] | | √ | | | |
| Wang et al. [27] | | √ | √ | | |
| Longo et al. [35] | √ | | | | |
| Schmidt et al. [36] | √ | | | | |
| Yi et al. [29] | | | √ | | √ |
| Wang et al. [25] | | √ | √ | | |
| Zhao et al. [31] | | | √ | | |
| Liu et al. [38] | √ | | | | |
| Bhatia et al. [40] | √ | | | √ | |
| Wang et al. [26] | | √ | √ | | |
| Zhang et al. [32] | | | √ | | √ |
| Zhang et al. [33] | | | √ | | |
| Camel et al. [34] | √ | | | | |
| Wu et al. [37] | √ | | | | |
| Sharma et al. [41] | √ | | | | |
| Babaei et al. [39] | √ | | | | |
| This paper | √ | √ | √ | √ | √ |

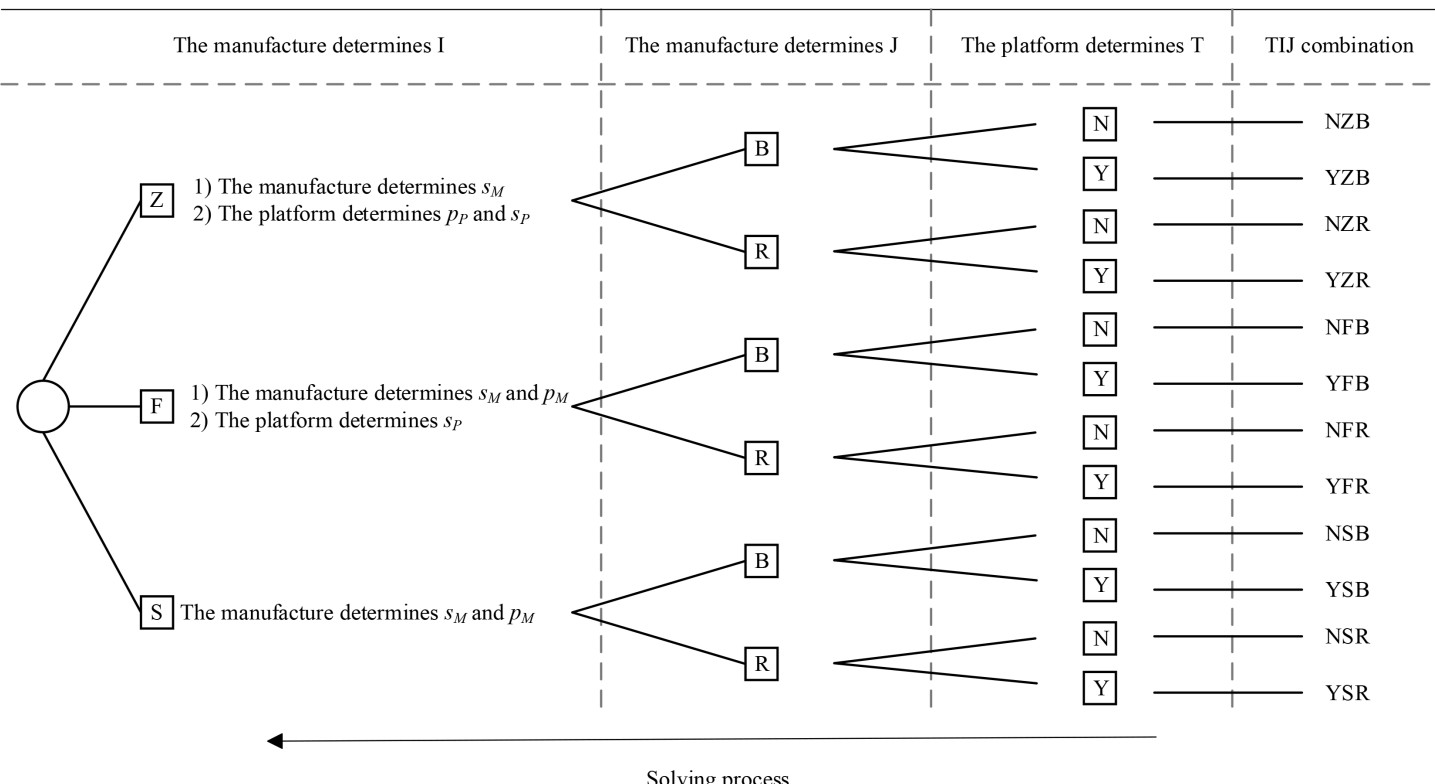

Fig 1. Sequence of events.

**Table 2. Meaning of model parameters.**

| Symbol | Description |
|---|---|
| Parameters | |
| $p_D$ | Direct channel sales price |
| $\alpha_I$ | Consumer preference for platform channels |
| $\gamma$ | Consumer sensitivity to the level of service |
| $k$ | Service cost factor |
| $c$ | Unit cost of production |
| $\omega$ | Wholesale price per unit |
| $\phi$ | Commission rate |
| $r_J$ | Loan interest rate and equity ratio |
| $U_j$ | Consumer utility |
| $Q_j$ | Number of channel sales |
| $Q$ | Total quantity sold |
| $\Pi$ | Manufacturer's profit |
| $\Omega$ | E-commerce platform profit |
| $\mu$ | Impact on platform channel preferences |
| $c_b$ | Blockchain unit application costs |
| $\theta$ | Spillover to direct sales channels |
| Decision variables | |
| $s_i$ | Product service level |
| $p_i$ | Platform channel sales price |

## 3.1 Model assumptions and notations

This section outlines the core assumptions underpinning our model. We provide justifications based on existing literature and common industry practices, and later discuss the robustness of our key findings to these assumptions in Section 3.3.

We assume a market of consumers who are heterogeneous in their valuation, $v$, for the product. Following a common approach in the operations management and marketing literature to model consumer heterogeneity with analytical tractability [26], we assume $v$ follows a uniform distribution on the interval $[0, 1]$, and the total market size is normalized to 1. A consumer's utility is influenced by the sales channel, product price, service level, and the presence of blockchain technology.

The valuation a consumer derives is channel-dependent. For a purchase through the manufacturer's direct channel, the valuation is $v$. For a purchase through the e-commerce platform, this valuation is adjusted by a preference parameter. As defined in Table 2, $\alpha_I$ represents the "Consumer preference for platform channels." Consequently, the valuation derived from the platform channel is $\alpha_I v$. We assume $\alpha_I > 1$, indicating that consumers inherently favor the platform channel due to factors like convenience or user experience, irrespective of the entry mode.

The adoption of blockchain technology further enhances the platform's appeal. By providing transparent and verifiable information about the product's supply chain journey, blockchain increases consumer trust. We model this effect using the parameter $\mu$, which Table 2 defines as the "Impact on platform channel preferences." When blockchain is implemented, the valuation consumers derive from the platform channel is elevated to $\mu\alpha_I v$. The condition $\mu > 1$ signifies that the verifiable transparency provided by blockchain adds a premium to the consumer's perceived value, thereby strengthening their preference for the platform.

In addition to product valuation, consumers also derive utility from the service level associated with their purchase. In the baseline scenario without blockchain, this utility is modeled as $\gamma s_i$. As defined in Table 2, $\gamma$ is the "Consumer sensitivity to the level of service" (with $0 < \gamma < 1$), and $s_i$ is the "Product service level" provided in channel $i$.

The introduction of blockchain technology enhances the perceived value of these services. When the platform adopts blockchain, consumers can verify the promised service levels, which increases their trust and serves a "regulatory role." This effect extends beyond the platform itself, creating a positive cross-channel impact. As the model posits, this is due to

factors like consumers' ability to access transparent information, which influences their perception of the manufacturer's overall service commitment.

This cross-channel effect is captured by the parameter $\theta$, which Table 2 defines as the "Spillover to direct sales channels." Consequently, even when purchasing from the manufacturer's direct channel, consumers perceive an enhanced service utility of $\theta\gamma s_M$. For model consistency, we assume this enhancement factor $\theta$ also applies to the platform channel, where the service verification is direct. Therefore, under the blockchain scenario, the service utility from any channel $i$ becomes $\theta\gamma s_i$. The model imposes the constraint $\gamma < \theta\gamma < 1$, which implies both a positive spillover effect ($\theta > 1$) and a bounded total sensitivity ($\theta\gamma < 1$).

Based on the preceding assumptions regarding product valuation and service utility, we now formulate the total utility functions for consumers purchasing from each channel.

The baseline model:

$$\begin{cases} U_Z = \alpha_I v - p_P + \gamma s_P \\ U_F = \alpha_I v - p_M + \gamma s_P \\ U_S = \alpha_I v - p_M + \gamma s_M \end{cases}, \tag{1}$$

Blockchain model:

$$\begin{cases} U_Z = \mu\alpha_I v - p_P + s_P \\ U_F = \mu\alpha_I v - p_M + s_P \\ U_S = \mu\alpha_I v - p_M + s_M \end{cases}, \tag{2}$$

The utility of consumers purchasing products in direct channels is modeled as: Baseline model:

$$U_D = v - p_D + \gamma s_M, \tag{3}$$

Blockchain model (where in SOP mode $\theta\gamma = 1$):

$$U_D = v - p_D + \theta\gamma s_M, \tag{4}$$

where $p_i$ and $p_D$ are respectively the price of the product on the platform channel and the direct sales channel. Consumers' platform purchase decision is based on utility $U_I$, while direct sales purchase decision is based on utility $U_D$. This paper defines $v_1$ and $v_2$ respectively as the indifferent values of $U_I = U_D$ and $U_D = 0$. Therefore, the product demand on the platform is $Q_I = 1 - v_1$, and the product demand on the direct channel is $Q_D = v_1 - v_2$. When the platform introduces a blockchain service, we model the implementation cost as an exogenous fixed cost for the platform, denoted as $c_b$. This represents the significant upfront investment required for the technology. Additionally, for each unit sold on the platform under the blockchain scenario, the manufacturer incurs a unit application cost $c_b$ (as defined in Table 2), which includes expenses related to data transaction and storage on the blockchain.

Providing a higher service level necessitates significant investment in areas such as logistics, technology, or personnel. It is widely recognized that achieving incremental improvements becomes progressively more costly as the service level rises. To capture this principle of increasing marginal costs, we model the cost of providing a service level $s_i$ using a convex, quadratic function $\frac{1}{2}ks_i^2$. In this formulation, k is the service cost coefficient, with $k > 0$.

### 3.2 Game sequence and financing

We model the strategic interactions between a manufacturer and an e-commerce platform across three distinct sales models. In the self-operated mode, the platform acts as a reseller. The manufacturer, as the Stackelberg leader, initially sets the service level $s_M$ for its direct sales channel. Subsequently, the platform purchases products from the manufacturer at a wholesale price $\omega$ and independently determines its retail price $p_P$ and service level $s_P$. In the other two modes, the agency models (FBP and SOP), the platform functions as a marketplace, enabling the manufacturer to sell directly to consumers. The manufacturer determines the product's selling price on the platform, while the platform charges a commission rate $\phi$ on sales. In these models, we assume the commission rate $\phi$ is an exogenous parameter. This is consistent with industry practice where major platforms set standard, non-negotiable rates for specific product categories, and is a common assumption in the platform economics literature [42]. The distinction between these two models lies in the responsibility for the platform channel's service level: it is determined by the platform in the FBP mode, whereas it is determined by the manufacturer in the SOP mode. Following the determination of market demand $q$, the manufacturer secures upfront production funds via one of two financing methods: 1) bank financing, which incurs an interest cost of $r_B cq$; or 2) equity financing, in which the manufacturer cedes a profit share of $r_R$. Finally, we consider the platform's strategic decision to adopt blockchain technology as a solution to information asymmetry. The platform evaluates the supply chain's operational performance under scenarios with and without blockchain to inform its implementation strategy.

### 3.3 Discussion of key model assumptions

In this section, we provide further justification for several key assumptions and discuss the potential implications of relaxing them, thereby assessing the robustness of our core findings.

1. Uniform Distribution of Consumer Valuation: As stated, our assumption that consumer valuation $v$ is uniformly distributed on $[0, 1]$ is a standard approach in the literature [26] that enables the derivation of a linear demand function and tractable, closed-form solutions.
   **Robustness:** This assumption is primarily made for analytical tractability. Our core qualitative insights are expected to be robust for other demand functions that are linear or near-linear. While a shift to a non-linear demand function would significantly complicate the analysis, we conjecture that the fundamental trade-offs we identify—such as the interplay between financing choices, blockchain costs, and operational modes—would persist, as they are not intrinsically tied to the specific shape of the demand curve.

2. Fixed Commission Rate: We assume the commission rate $\phi$ is an exogenous parameter. As justified earlier, this reflects real-world platform pricing strategies and is a common simplification in the literature [42,43].
   **Robustness:** Endogenizing the commission rate would introduce an additional strategic layer. The platform might then adjust the rate to either attract more manufacturers or extract more rent. While the quantitative results would change, we expect our central qualitative finding—that the platform's blockchain adoption decision is contingent on the manufacturer's financing mode—would still hold, as these fundamental interactions would continue to shape the ecosystem's profitability.

3. Exogenous Blockchain Cost: We model the blockchain implementation cost as an exogenous fixed cost, $c_b$, borne by the platform. This simplification is common in models of technology adoption [44] and represents the significant upfront investment required.
   **Robustness:** An alternative would be to model this as a variable cost dependent on transaction volume. Such a change would influence the optimal pricing and service levels but would not fundamentally alter the strategic adoption decision itself. This decision is driven by the overarching trade-off between the total cost of the technology and its benefits (captured by $\mu$ and $\theta$). Therefore, we expect our core insights regarding the adoption thresholds to be robust to this modification.

## 3.4 Analytical framework

We structure our analysis around two primary scenarios based on the platform's blockchain adoption strategy: a scenario without blockchain implementation (denoted by $N$) and one with it (denoted by $Y$). Within each scenario, we examine the strategic interactions under three entry modes ($I$): a self-operated mode ($Z$), a FBP mode ($F$), and a SOP mode ($S$). Furthermore, for each of these combinations, we consider the two financing modes ($J$) available to the manufacturer: bank financing ($B$) and equity financing ($R$). For clarity and conciseness, each specific model combination is identified using the composite notation $TIJ$, where $T \in \{N, Y\}$, $I \in \{Z, F, S\}$, and $J \in \{B, R\}$. Throughout the paper, the subscripts $i = \{P, M\}$ are used to denote the players (e-commerce platform and manufacturer, respectively), and the subscripts $j = \{I, D\}$ are used to denote the sales channels (platform channel and direct sales channel, respectively).

The overall dual-channel supply chain structure for each financing mode is depicted in Figs 2 and 3, respectively. The detailed decision sequences for each game are illustrated in Fig 1.

In the subsequent sections, we apply Stackelberg game theory and use backward induction to solve for the equilibrium strategies for the manufacturer and the e-commerce platform under each of these twelve model combinations.

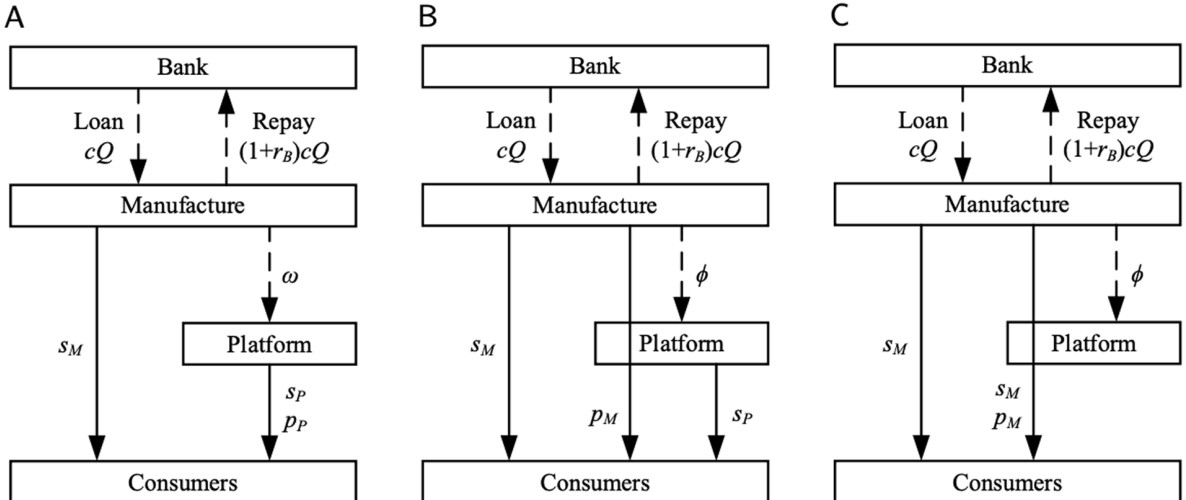

**Fig 2. Dual-channel structure under bank financing.** (A) Self-operated mode. (B) FBP mode. (C) SOP mode.

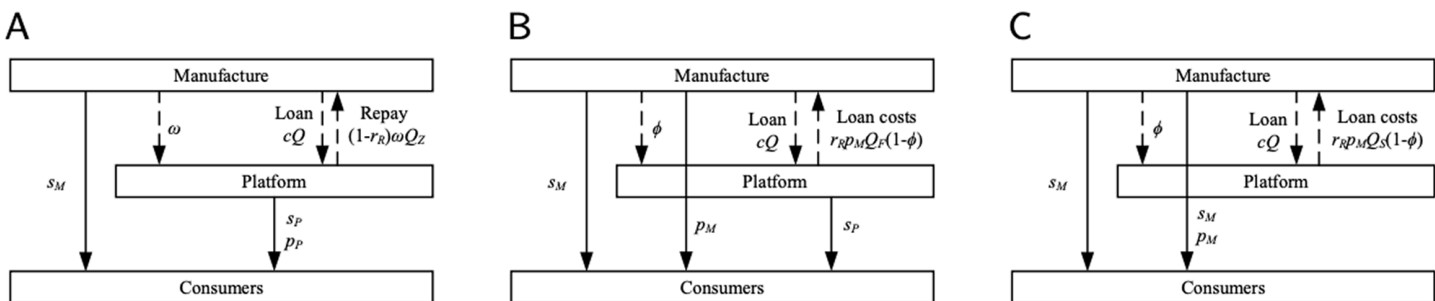

**Fig 3. Dual-channel structure under equity financing.** (A) Self-operated mode. (B) FBP mode. (C) SOP mode.

# 4 Model analysis and equilibrium solutions

## 4.1 Benchmark model

In this section, we analyze the equilibrium outcomes for the benchmark models, which represent all scenarios without the implementation of blockchain technology. Specifically, we solve for the equilibrium in the six model combinations: NZB, NZR, NFB, NFR, NSB, and NSR.

**4.1.1 Self-operated mode.** In the self-operated mode, the interaction follows a Stackelberg game structure. The manufacturer, as the leader, first determines the service level of the direct sales channel ($s_M$). Subsequently, the platform, as the follower, decides on the service level ($s_P$) and the selling price ($p_P$) for the platform channel.

The utility functions for consumers purchasing from the platform channel and the direct sales channel are formulated as follows:

$$\begin{cases} U_Z = \alpha_Z v - p_P + \gamma s_P \\ U_D = v - p_D + \gamma s_M \end{cases}, \tag{5}$$

We define $v_1$ and $v_2$ respectively as the indifference values of $U_Z = U_D$ and $U_D = 0$. Under that value, getting the $v_1 = \frac{p_P - p_D + \gamma(s_M - s_P)}{\alpha_Z - 1}$ and $v_2 = p_D - \gamma s_M$, respectively. Therefore, the product demand of the platform channel is $Q_Z = 1 - v_1 = 1 + \frac{p_P - p_D + \gamma(s_P - s_M)}{\alpha_Z - 1}$, the product demand of the direct channel is $Q_D = v_1 - v_2 = \frac{p_P - \alpha_Z p_D + \gamma(\alpha_Z s_M - s_P)}{\alpha_Z - 1}$, and the total market demand is $Q = Q_Z + Q_D$.

**1. Bank financing** The profit functions of the manufacturer and the e-commerce platform are:

$$\begin{cases} \Pi^{NZB} = \omega Q_Z + p_D Q_D - \frac{1}{2} k s_M^2 - cQ(1 + r_B) \\ \Omega^{NZB} = (p_P - \omega) Q_Z - \frac{1}{2} k s_P^2 \end{cases}, \tag{6}$$

**Proposition 1.** *When $\alpha_Z > 1 + \frac{\gamma^2}{2k}$, there have optimal solutions.*
*The e-commerce decision variables are:*

$$\begin{cases} p_P^{NZB*} = \omega + \frac{m_1 g_1 k}{l_1} \\ s_P^{NZB*} = \frac{\gamma g_1}{l_1} \end{cases}, \tag{7}$$

*The manufacturer decision variable is:*

$$s_M^{NZB*} = \gamma h_1, \tag{8}$$

*where $m_1 = \alpha_Z - 1$, $l_1 = 2m_1 k - \gamma^2$, $n_1 = 1 + r_B$, $h_1 = \frac{p_D - \omega}{l_1} + \frac{p_D - c n_1}{k}$, $g_1 = m_1 + p_D - \omega - \gamma^2 h_1$.*

**Proof Sketch:** The proof follows a two-stage backward induction approach. First, we analyze the e-commerce platform's decision. We establish that the platform's profit function is jointly concave in its selling price ($p_P$) and service level ($s_P$) by verifying that its Hessian matrix is negative definite under the condition $\alpha_Z > 1 + \frac{\gamma^2}{2k}$. We then derive the platform's optimal response functions using the first-order conditions. Second, we substitute these response functions into the manufacturer's profit function. We show that the manufacturer's profit function is concave in its own service level ($s_M$) and solve for the optimal $s_M$. Finally, substituting the manufacturer's optimal decision back into the platform's response functions yields the unique equilibrium solutions. The detailed derivation process of all proofs is included in the S1 Appendix.

**2. Equity financing** The profit functions of the manufacturer and the e-commerce platform are:

$$\begin{cases} \Pi^{NZR} = (1 - r_R)\omega Q_Z + p_D Q_D - \frac{1}{2}ks_M^2 - cQ \\ \Omega^{NZR} = (p_P - \omega)Q_Z - \frac{1}{2}ks_P^2 + r_R\omega Q_Z \end{cases}, \tag{9}$$

**Proposition 2.** *When $\alpha_Z > 1 + \frac{\gamma^2}{2k}$, there have optimal solutions.*
*The e-commerce decision variables are:*

$$\begin{cases} p_P^{NZR*} = n_2\omega + \frac{m_1 g_2 k}{l_1} \\ s_P^{NZR*} = \frac{\gamma g_2}{l_1} \end{cases}, \tag{10}$$

*The manufacturer decision variable is:*

$$s_M^{NZR*} = \gamma h_2, \tag{11}$$

where $n_2 = 1 - r_R$, $h_2 = \frac{p_D - n_2\omega}{l_1} + \frac{p_D - c}{k}$, $g_2 = m_1 + p_D - n_2\omega - \gamma^2 h_2$.

**4.1.2 FBP mode.** Under the FBP mode, which is also a Stackelberg game, the manufacturer (leader) first determines the service level ($s_M$) and selling price ($p_M$) for its direct sales channel. The platform (follower) then decides the service level ($s_P$) for the platform channel.

The consumer utility functions for the platform and direct sales channels are specified as follows:

$$\begin{cases} U_F = \alpha_F v - p_M + \gamma s_P \\ U_D = v - p_D + \gamma s_M \end{cases}, \tag{12}$$

We define $v_1$ and $v_2$ respectively as $U_F = U_D$ the indifference values of and $U_D = 0$. Under that value, getting the $v_1 = \frac{p_M - p_D + \gamma(s_M - s_P)}{\alpha_F - 1}$ and $v_2 = p_D - \gamma s_M$, respectively. Therefore, the product demand of the platform channel is $Q_F = 1 - v_1 = 1 + \frac{p_D - p_M + \gamma(s_P - s_M)}{\alpha_F - 1}$, the product demand of the direct channel is $Q_D = v_1 - v_2 = \frac{p_M - \alpha_F p_D + \gamma(\alpha_F s_M - s_P)}{\alpha_F - 1}$, and the total market demand is $Q = Q_F + Q_D$.

**1. Bank financing** The profit functions of the manufacturer and the e-commerce platform are:

$$\begin{cases} \Pi^{NFB} = (1 - \phi)p_M Q_F + p_D Q_D - \frac{1}{2}ks_M^2 - cQ(1 + r_B) \\ \Omega^{NFB} = \phi p_M Q_F - \frac{1}{2}ks_P^2 \end{cases}, \tag{13}$$

**Proposition 3.** *When $\alpha_F > 1 + \frac{\gamma^2(1 + \phi)}{2k}$, there have optimal solutions.*
*The e-commerce decision variable is:*

$$s_P^{NFB*} = \frac{\phi\gamma h_3}{kl_2}, \tag{14}$$

*The manufacturer decision variables are:*

$$\begin{cases} s_M^{NFB*} = \frac{\gamma g_3}{k} \\ p_M^{NFB*} = \frac{m_1 h_3}{l_2} \end{cases},$$ (15)

*where* $n_3 = 1 - \phi$, $l_2 = 2m_1 k - \gamma^2(1 + \phi)$, $h_3 = k(m_1 + p_D) - \gamma^2(p_D - cn_1) + \frac{p_D(m_1 k - \gamma^2)}{n_3 m_1}$, $g_3 = \frac{p_D}{m_1} + p_D - cn_1 - \frac{n_3 h_3}{l_2}$.

**Proof Sketch:** The proof is based on a two-stage backward induction process. In the first stage, we analyze the e-commerce platform's decision on its service level ($s_P$). We demonstrate that the platform's profit function is concave in $s_P$ by examining its second-order derivative, and we derive its optimal response function. In the second stage, we address the manufacturer's problem. By substituting the platform's optimal response into the manufacturer's profit function, we establish that this function is jointly concave in the manufacturer's own service level ($s_M$) and selling price ($p_M$) under the condition $\alpha_F > 1 + \frac{\gamma^2(1+\phi)}{2k}$. This is confirmed by verifying the negative definiteness of the corresponding Hessian matrix. We then simultaneously solve for the optimal $s_M$ and $p_M$ using the first-order conditions. Finally, substituting the manufacturer's optimal price back into the platform's response function yields the equilibrium service level for the platform.

**2. Equity financing** The profit functions of the manufacturer and the e-commerce platform are:

$$\begin{cases} \Pi^{NFR} = (1 - r_R)(1 - \phi)p_M Q_F + p_D Q_D - \frac{1}{2}ks_M^2 - cQ \\ \Omega^{NFR} = \phi p_M Q_F - \frac{1}{2}ks_P^2 + r_R(1 - \phi)p_M Q_F \end{cases},$$ (16)

**Proposition 4.** *When* $\alpha_F > 1 + \frac{\gamma^2(2 - n_2 n_3)}{2k}$, *there have optimal solutions.*
*The e-commerce decision variable is:*

$$s_P^{NFR*} = \frac{\gamma h_4(1 - n_2 n_3)}{kl_3},$$ (17)

*The manufacturer decision variables are:*

$$\begin{cases} s_M^{NFR*} = \frac{\gamma g_4}{k} \\ p_M^{NFR*} = \frac{m_1 h_4}{l_3} \end{cases},$$ (18)

*where* $l_3 = 2m_1 k - \gamma^2(2 - n_2 n_3)$, $h_4 = k(m_1 + p_D) - \gamma^2(p_D - c) + \frac{p_D(m_1 k - \gamma^2)}{n_2 n_3 m_1}$, $g_4 = \frac{p_D}{m_1} + p_D - c - \frac{n_2 n_3 h_4}{l_3}$.

**4.1.3 SOP mode.** In the SOP mode, the manufacturer is the sole decision-maker, determining both the service level ($s_M$) and the selling price ($p_M$) for the direct sales channel.

The resulting consumer utility functions for the platform and direct sales channels are:

$$\begin{cases} U_S = \alpha_S v - p_M + \gamma s_M \\ U_D = v - p_D + \gamma s_M \end{cases},$$ (19)

We define define $v_1$ and $v_2$ respectively as the indifferent values of $U_S = U_D$ and $U_D = 0$. Under that value, getting the $v_1 = \frac{p_M - p_D}{\alpha_S - 1}$ and $v_2 = p_D - \gamma s_M$, respectively. Therefore, the product demand of the platform channel is $Q_S = 1 - v_1 = 1$

$+ \frac{p_D - p_M}{\alpha_S - 1}$, the product demand of the direct channel is $Q_D = v_1 - v_2 = \gamma s_M + \frac{p_M - \alpha_S p_D}{\alpha_S - 1}$, and the total market demand is $Q = Q_S + Q_D$.

**1. Bank financing** The profit functions of the manufacturer and the e-commerce platform are:

$$\begin{cases} \Pi^{NSB} = (1 - \phi)p_M Q_S + p_D Q_D - k s_M^2 - cQ(1 + r_B) \\ \Omega^{NSB} = \phi p_M Q_S \end{cases} , \tag{20}$$

**Proposition 5.** *When $\alpha_S > 1$, there have optimal solutions.*
*The manufacturer decision variables are:*

$$\begin{cases} s_M^{NSB*} = \frac{\gamma(p_D - cn_1)}{2k} \\ p_M^{NSB*} = \frac{h_5}{2} \end{cases} , \tag{21}$$

*where $h_5 = m_1 + p_D + \frac{p_D}{n_3}$.*

**Proof Sketch:** In this scenario, the problem reduces to a single optimization problem for the manufacturer, who simultaneously determines the service level ($s_M$) and the selling price ($p_M$). We begin by establishing the joint concavity of the manufacturer's profit function. This is accomplished by showing that its Hessian matrix is negative definite, a condition that holds for all $\alpha_S > 1$. A key feature of this model is that the Hessian matrix is diagonal, which indicates that the cross-partial derivatives are zero and the decisions on service and price are separable at the second order. Consequently, the unique optimal solutions for $s_M$ and $p_M$ can be found directly by solving their respective first-order conditions.

**2. Equity financing** The profit functions of the manufacturer and the e-commerce platform are:

$$\begin{cases} \Pi^{NSR} = (1 - r_R)(1 - \phi)p_M Q_S + p_D Q_D - k s_M^2 - cQ \\ \Omega^{NSR} = \phi p_M Q_S + r_R(1 - \phi)p_M Q_S \end{cases} , \tag{22}$$

**Proposition 6.** *When $\alpha_S > 1$, there have optimal solutions.*
*The manufacturer decision variables are:*

$$\begin{cases} s_M^{NSR*} = \frac{\gamma(p_D - cn_1)}{2k} \\ p_M^{NSB*} = \frac{h_6}{2} \end{cases} , \tag{23}$$

*where $h_6 = m_1 + p_D + \frac{p_D}{n_2 n_3}$.*

Table 3 presents the equilibrium solutions for the six model combinations in the absence of blockchain services.

## 4.2 Blockchain model

We now analyze the scenarios in which the platform implements blockchain technology. The primary modeling assumption is that blockchain adoption directly impacts the consumer utility function. Accordingly, we solve for the equilibrium in the six corresponding model combinations: YZB, YZR, YFB, YFR, YSB, and YSR. The decision sequences for these models are identical to those of their benchmark counterparts; therefore, we employ the same backward induction method for the analysis.

**Table 3**. Equilibrium results under different models without considering blockchain services.

| Entry Mode | Parameters | Bank financin | Equity financin |
|---|---|---|---|
| Self-operated Mode | $p_P$ | $\omega + \frac{m_1 g_1 k}{l_1}$ | $n_2\omega + \frac{m_1 g_2 k}{l_1}$ |
| | $s_P$ | $\frac{\gamma g_1}{l_1}$ | $\frac{\gamma g_2}{l_1}$ |
| | $s_M$ | $\gamma h_1$ | $\gamma h_2$ |
| FBP Mode | $s_P$ | $\frac{\phi\gamma h_3}{kl_2}$ | $\frac{\gamma h_4(1-n_2 n_3)}{kl_3}$ |
| | $s_M$ | $\frac{\gamma g_3}{k}$ | $\frac{\gamma g_4}{k}$ |
| | $p_M$ | $\frac{m_1 h_3}{l_2}$ | $\frac{m_1 h_4}{l_3}$ |
| SOP Mode | $s_M$ | $\frac{\gamma(p_D-cn_1)}{2k}$ | $\frac{\gamma(p_D-cn_1)}{2k}$ |
| | $p_M$ | $\frac{h_5}{2}$ | $\frac{h_6}{2}$ |

**4.2.1 Self-operated mode.** The consumer utility functions of platform channel and direct marketing channel are as follows:

$$\begin{cases} U_Z = \mu\alpha_Z v - p_P + s_P \\ U_D = v - p_D + \theta\gamma s_M \end{cases}, \tag{24}$$

We define $v_1$ and $v_2$ respectively as the indifference values of $U_Z = U_D$ and $U_D = 0$. Under that value, getting the $v_1 = \frac{p_P - p_D + \theta\gamma s_M - s_P}{\mu\alpha_Z - 1}$ and $v_2 = p_D - \theta\gamma s_M$, respectively. Therefore, the product demand of the platform channel is $Q_Z = 1 - v_1 = 1 + \frac{p_D - p_P + s_P - \theta\gamma s_M}{\mu\alpha_Z - 1}$, the product demand of the direct channel is $Q_D = v_1 - v_2 = \frac{p_P - \nu\alpha_Z p_D + \mu\theta\gamma\alpha_Z s_M - s_P}{\mu\alpha_Z - 1}$, and the total market demand is $Q = Q_Z + Q_D$.

**1. Bank financing** The profit functions of the manufacturer and the e-commerce platform are:

$$\begin{cases} \Pi^{YZB} = \omega Q_Z + p_D Q_D - \frac{1}{2}ks_M^2 - cQ(1+r_B) - c_b Q_Z \\ \Omega^{YZB} = (p_P - \omega)Q_Z - \frac{1}{2}ks_P^2 + c_b Q_Z \end{cases}, \tag{25}$$

**Proposition 7.** When $\alpha_Z > \frac{1}{\mu}(1 + \frac{1}{2k})$, there have optimal solutions.
*The e-commerce decision variables are:*

$$\begin{cases} p_P^{YZB*} = \omega - c_b + \frac{m_2 G_1 k}{L_1} \\ s_P^{YZB*} = \frac{G_1}{L_1} \end{cases}, \tag{26}$$

*The manufacturer decision variable is:*

$$s_M^{YZB*} = \theta\gamma H_1, \tag{27}$$

*where $m_2 = \mu\alpha_Z - 1$, $L_1 = 2m_2 k - 1$, $H_1 = \frac{c_b + p_D - \omega}{L_1} + \frac{p_D - cn_1}{k}$, $G_1 = c_b + m_2 + p_D - \omega - \theta^2\gamma^2 H_1$.*

**2. Equity financing** The profit functions of the manufacturer and the e-commerce platform are:

$$\begin{cases} \Pi^{YZR} = (1 - r_R)\omega Q_Z + p_D Q_D - \frac{1}{2}ks_M^2 - cQ - c_b Q_Z \\ \Omega^{YZR} = (p_P - \omega)Q_Z - \frac{1}{2}ks_P^2 + r_R\omega Q_Z + c_b Q_Z \end{cases}, \tag{28}$$

**Proposition 8.** When $\alpha_Z > \frac{1}{\mu}(1 + \frac{1}{2k})$, there have optimal solutions.
The e-commerce decision variables are:

$$\begin{cases} p_P^{YZR*} = n_2\omega - c_b + \frac{m_2 G_2 k}{L_1} \\ s_P^{YZR*} = \frac{G_2}{L_1} \end{cases}, \tag{29}$$

The manufacturer decision variable is:

$$s_M^{YZR*} = \theta\gamma H_2, \tag{30}$$

where $H_2 = \frac{c_b + p_D - n_2\omega}{L_1} + \frac{p_D - c}{k}$, $G_2 = c_b + m_2 + p_D - n_2\omega - \theta^2\gamma^2 H_2$.

### 4.2.2 FBP mode.

The consumer utility functions of the platform channel and the direct marketing channel are as follows:

$$\begin{cases} U_F = \mu\alpha_F v - p_M + s_P \\ U_D = v - p_D + \theta\gamma s_M \end{cases}, \tag{31}$$

We define $v_1$ and $v_2$ respectively as $U_F = U_D$ the indifference values of and $U_D = 0$. Under that value, getting the $v_1 = \frac{p_M - p_D + \theta\gamma s_M - s_P}{\mu\alpha_F - 1}$ and $v_2 = p_D - \theta\gamma s_M$, respectively. Therefore, the product demand of the platform channel is $Q_F = 1 - v_1 = 1 + \frac{p_D - p_M + s_P - \theta\gamma s_M}{\mu\alpha_F - 1}$, the product demand of the direct channel is $Q_D = v_1 - v_2 = \frac{p_M - \mu\alpha_F p_D + \mu\theta\gamma\alpha_F s_M - s_P}{\mu\alpha_F - 1}$, and the total market demand is $Q = Q_F + Q_D$.

**1. Bank financing** The profit functions of the manufacturer and the e-commerce platform are:

$$\begin{cases} \Pi^{YFB} = (1 - \phi)p_M Q_F + p_D Q_D - \frac{1}{2}ks_M^2 - cQ(1 + r_B) - c_b Q_F \\ \Omega^{YFB} = \phi p_M Q_F - \frac{1}{2}ks_P^2 + c_b Q_F \end{cases}, \tag{32}$$

**Proposition 9.** When $\alpha_F > \frac{1}{\mu}(1 + \frac{n_3\theta^2\gamma^2 + 2\phi}{2k})$, there have optimal solutions.
The e-commerce decision variable is:

$$s_P^{YFB*} = \frac{c_b + \phi H_3}{m_2 k}, \tag{33}$$

The manufacturer decision variables are:

$$\begin{cases} s_M^{YFB*} = \frac{\theta\gamma G_3}{k} \\ p_M^{YFB*} = H_3 \end{cases}, \tag{34}$$

where $L_2 = 2m_2k - n_3\theta^2\gamma^2 - 2\phi$, $e_1 = k(m_2 + p_D) - \theta^2\gamma^2(p_D - cn_1) + \frac{(c_b + p_D)(1 - m_2k)}{n_3m_2}$, $H_3 = \frac{c_b + p_D}{n_3} + \frac{e_1m_2 - p_D}{L_2}$, $G_3 = p_D - cn_1 - \frac{n_3(e_1m_2 - p_D)}{m_2L_2}$.

**2. Equity financing** The profit functions of the manufacturer and the e-commerce platform are:

$$\begin{cases} \Pi^{YFR} = (1 - r_R)(1 - \phi)p_MQ_F + p_DQ_D - \frac{1}{2}ks_M^2 - cQ - c_bQ_F \\ \Omega^{YFR} = \phi p_MQ_F - \frac{1}{2}ks_P^2 + r_R(1 - \phi)p_MQ_F + c_bQ_F \end{cases}, \tag{35}$$

**Proposition 10.** *When $\alpha_F > \frac{1}{\mu}(1 + \frac{n_2n_3\theta^2\gamma^2 + 2(1 - n_2n_3)}{2k})$, there have optimal solutions.*
*The e-commerce decision variable is:*

$$s_P^{YFR*} = \frac{c_b + H_4(1 - n_2n_3)}{m_2k}, \tag{36}$$

*The manufacturer decision variables are:*

$$\begin{cases} s_M^{YFR*} = \frac{\theta\gamma G_4}{k} \\ p_M^{YFR*} = H_4 \end{cases}, \tag{37}$$

where $L_3 = 2m_2k - n_2n_3\theta^2\gamma^2 - 2(1 - n_2n_3)$, $e_2 = k(m_2 + p_D) - \theta^2\gamma^2(p_D - c) + \frac{(c_b + p_D)(2 - \theta^2\gamma^2)}{2m_2}$, $H_4 = \frac{c_b + p_D}{2n_2n_3} + \frac{e_2m_2 - p_D}{L_3}$, $G_4 = p_D - cn_1 - \frac{n_2n_3(e_2m_2 - p_D)}{m_2L_3}$.

#### 4.2.3 SOP mode.
The consumer utility functions of the platform channel and the direct selling channel are:

$$\begin{cases} U_S = \mu\alpha_S v - p_M + s_M \\ U_D = v - p_D + s_M \end{cases}, \tag{38}$$

We define $v_1$ and $v_2$ respectively as the indifferent values of $U_S = U_D$ and $U_D = 0$. Under that value, getting the $v_1 = \frac{p_M - p_D}{\mu\alpha_S - 1}$ and $v_2 = p_D - s_M$, respectively. Therefore, the product demand of the platform channel is $Q_S = 1 - v_1 = 1 + \frac{p_D - p_M}{\mu\alpha_S - 1}$, the product demand of the direct channel is $Q_D = v_1 - v_2 = s_M + \frac{p_M - \mu\alpha_S p_D}{\mu\alpha_S - 1}$, and the total market demand is $Q = Q_S + Q_D$.

**1. Bank financing** The profit functions of the manufacturer and the e-commerce platform are:

$$\begin{cases} \Pi^{YSB} = (1 - \phi)p_MQ_S + p_DQ_D - ks_M^2 - cQ(1 + r_B) - c_bQ_S \\ \Omega^{YSB} = \phi p_MQ_S + c_bQ_S \end{cases}, \tag{39}$$

**Proposition 11.** *When $\alpha_S > \frac{1}{\mu}$, there have optimal solutions.*
*The manufacturer decision variables are:*

$$\begin{cases} s_M^{YSB*} = \frac{p_D - cn_1}{2k} \\ p_M^{YSB*} = \frac{H_5}{2} \end{cases}, \tag{40}$$

where $H_5 = m_2 + p_D + \frac{c_b + p_D}{n_3}$.

**2. Equity financing** The profit functions of the manufacturer and the e-commerce platform are:

$$\begin{cases} \Pi^{YSR} = (1 - r_R)(1 - \phi)p_M Q_S + p_D Q_D - k s_M^2 - cQ - c_b Q_S \\ \Omega^{YSR} = \phi p_M Q_S + r_R(1 - \phi)p_M Q_S + c_b Q_S \end{cases}, \tag{41}$$

**Proposition 12.** *When $\alpha_S > \frac{1}{\mu}$, there have optimal solutions.*
*The manufacturer decision variables are:*

$$\begin{cases} s_M^{YSR*} = \frac{p_D - c}{2k} \\ p_M^{YSB*} = \frac{H_6}{2} \end{cases}, \tag{42}$$

*where $H_6 = m_2 + p_D + \frac{c_b + p_D}{n_2 n_3}$.*

Table 4 summarizes the equilibrium solutions for the six model combinations when blockchain services are integrated.

## 5 Numerical analysis

In this section, we conduct a numerical analysis to validate our theoretical findings and derive further managerial insights. The analysis is twofold. First, we perform a sensitivity analysis to investigate the impact of critical parameters on the optimal decisions and profits for each supply chain member across the different models. Second, we numerically compare the equilibrium outcomes of the benchmark models (without blockchain) with their blockchain-enabled counterparts. This comparison allows us to quantify the value of blockchain adoption and understand its strategic implications for the manufacturer and the platform. The parameter values are determined based on a combination of [26,45,46]. The following basic parameters are set in this paper: $r_B = 0.05$, $r_R = 0.03$, $\phi = 0.12$, $\alpha_Z = 1.5$, $\alpha_F = 1.4$, $\alpha_S = 1.3$, $k = 1$, $p_D = 0.55$, $c = 0.01$, $\omega = 0.35$, $\gamma = 0.2$, $c_b = 0.02$. When studying the impact of blockchain technology on channel preferences, in order to eliminate the influence of the spillover effects of blockchain technology, we set $\theta = 1.5$. When analyzing the spillover effects of blockchain technology, in order to eliminate the influence of the impact of blockchain technology on channel preferences, we set $\mu = 1.5$. In the analysis of Sects 5.2 and 5.3, we set $\theta = 2$, $\mu = 2$.

**Table 4**. Equilibrium results under different models considering blockchain services.

| Model | Parameters | Bank financin | Equity financin |
|---|---|---|---|
| Self-operated Mode | $p_P$ | $\omega - c_b + \frac{m_2 G_1 k}{L_1}$ | $n_2 \omega - c_b + \frac{m_2 G_2 k}{L_1}$ |
| | $s_P$ | $\frac{G_1}{L_1}$ | $\frac{G_2}{L_1}$ |
| | $s_M$ | $\theta \gamma H_1$ | $\theta \gamma H_2$ |
| FBP Model | $s_P$ | $\frac{c_b + \phi H_3}{m_2 k}$ | $\frac{c_b + H_4(1 - n_2 n_3)}{m_2 k}$ |
| | $s_M$ | $\frac{\theta \gamma G_3}{k}$ | $\frac{\theta \gamma G_4}{k}$ |
| | $p_M$ | $H_3$ | $H_4$ |
| SOP Model | $s_M$ | $\frac{p_D - c n_1}{2k}$ | $\frac{p_D - c}{2k}$ |
| | $p_M$ | $\frac{H_5}{2}$ | $\frac{H_6}{2}$ |

### 5.1 Sensitivity analysis of decision variables under the blockchain model

**5.1.1 The impact of blockchain technology on channel preferences.** To investigate how the increasing consumer preference for blockchain platforms affects the service level of traditional e-commerce platforms, we conducted a modeling analysis of the self-operated and FBP modes. Fig 4A illustrates the primary trend of this impact. As the figure shows, an increase in consumer preference for blockchain corresponds with a significant monotonic decrease in the platform's service level in both models. This outcome is influenced by factors such as fund liquidity and resource allocation. In the self-operated mode, where the platform directly manages sales and logistics, a significant capital investment is required. As consumer preference shifts toward blockchain, the competitiveness of the traditional platform may decline. The decentralized nature of blockchain, which enhances transparency and reduces transaction costs, exerts considerable pressure on the platform's capital requirements. To adapt, the platform might scale back non-core services or optimize existing structures, potentially leading to a lower overall service quality. In the FBP mode, while outsourcing alleviates some financial pressures, the platform remains vulnerable to shifts in market demand. As consumers and merchants divert transaction traffic to decentralized alternatives, the platform's market share and financing capabilities are reduced. To maintain competitiveness, the platform may be forced to adopt more cautious resource allocation strategies, resulting in diminished service levels, particularly in supply chain management and customer service.

Fig 4B illustrates how the manufacturer's service level changes with increasing consumer blockchain preference across three modes: self-operated, FBP, and SOP. In the self-operated mode, the service level shows a continuous downward trend. In contrast, under the FBP mode, the manufacturer's service level increases as consumer preference grows. The service level in the SOP mode remains nearly constant, indicating strong stability. These variations are driven by the platform's operating model and financing methods, reflecting different responses to cost control, competitive strategies, and market demand. In the self-operated mode, the rise of blockchain platforms heightens competitive pressure. To reduce operating costs, the platform may cut services by lowering support for manufacturers, which results in a decline in the manufacturer's service level, particularly under external financing where profit-driven cost reductions are more pronounced. In the FBP mode, the platform responds more flexibly to market demand. The growth of blockchain platforms can lead to an improvement in the manufacturer's service level, as the platform strengthens support for manufacturers to maintain market share. This is especially true under internal financing, which provides greater financial flexibility to invest in long-term competitive advantages. Consequently, the manufacturer's service level increases significantly under this financing mode. In the SOP mode, standardized processes make large-scale changes difficult in the short term. Therefore, the manufacturer's service level remains stable, with adjustments focusing on efficiency rather than significant service alterations.

Beyond service levels, the shift in consumer preferences also has a significant impact on the pricing strategies of e-commerce platforms. Fig 4C presents the dynamic trend of product sales prices under the three modes. The self-operated mode exhibits a "U-shaped" price trajectory. The FBP mode shows a robust, monotonically rising price, while the SOP mode displays a smoother, more stable upward trend. These trends are closely linked to the platform's financing, market competition, and cost structure. In the self-operated mode, heightened competition initially prompts the platform to lower prices to maintain market share. However, as supply chain and operational costs rise, the platform may struggle with low-price competition and subsequently raise prices to restore profitability, forming the U-shaped curve. In the FBP mode, outsourcing operating costs provides greater flexibility. As blockchain platforms gain popularity, the e-commerce platform may invest in supply chain optimization or value-added services to remain competitive, leading to higher prices to maintain profit margins. In the SOP mode, which follows standardized processes, the platform focuses on improving efficiency and reducing unnecessary costs. In this context, price increases are often linked to value-added improvements in areas like after-sales or logistics services. Due to the standardized nature of its operations, the price trends in the SOP mode tend to be more stable.

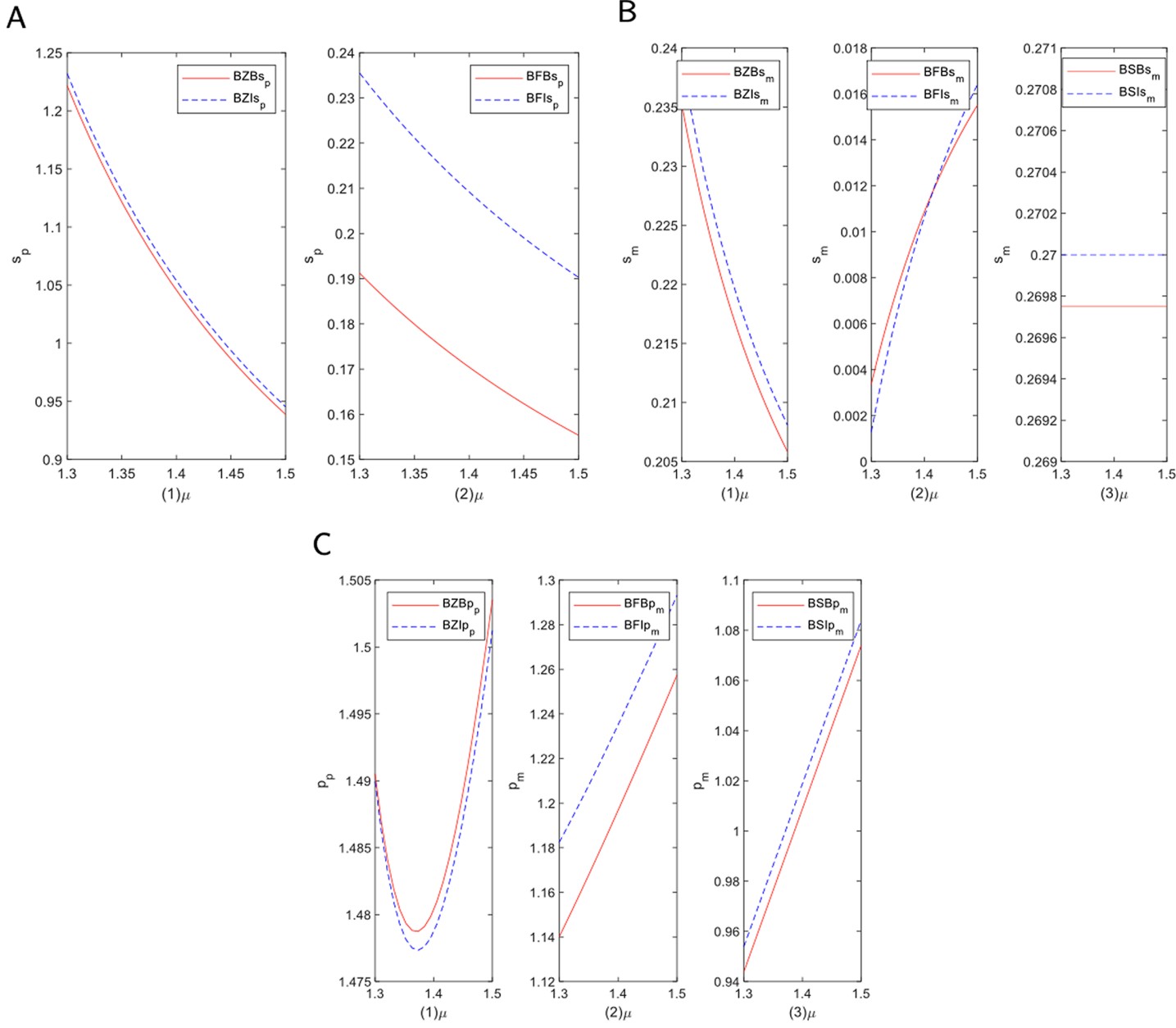

**Fig 4. Analysis of the impact of blockchain technology on channel preferences.** (A) E-commerce service level. (B) Manufacturer service level. (C) Platform selling price.

**5.1.2 Spillover effects of blockchain technology.** Fig 5A depicts how the spillover effects of blockchain technology dynamically alter the service level of e-commerce platforms under different modes. The figure shows that as the spillover effect of blockchain increases, the service levels of both the self-operated and FBP modes trend downward, though with significant differences in the rate and magnitude of their decline. The self-operated mode exhibits a steep downward trajectory, whereas the FBP mode's curve is much smoother. The spillover effect refers to the broad impact of blockchain technology on the platform and its surrounding ecosystem. In the self-operated mode, where the platform manages the

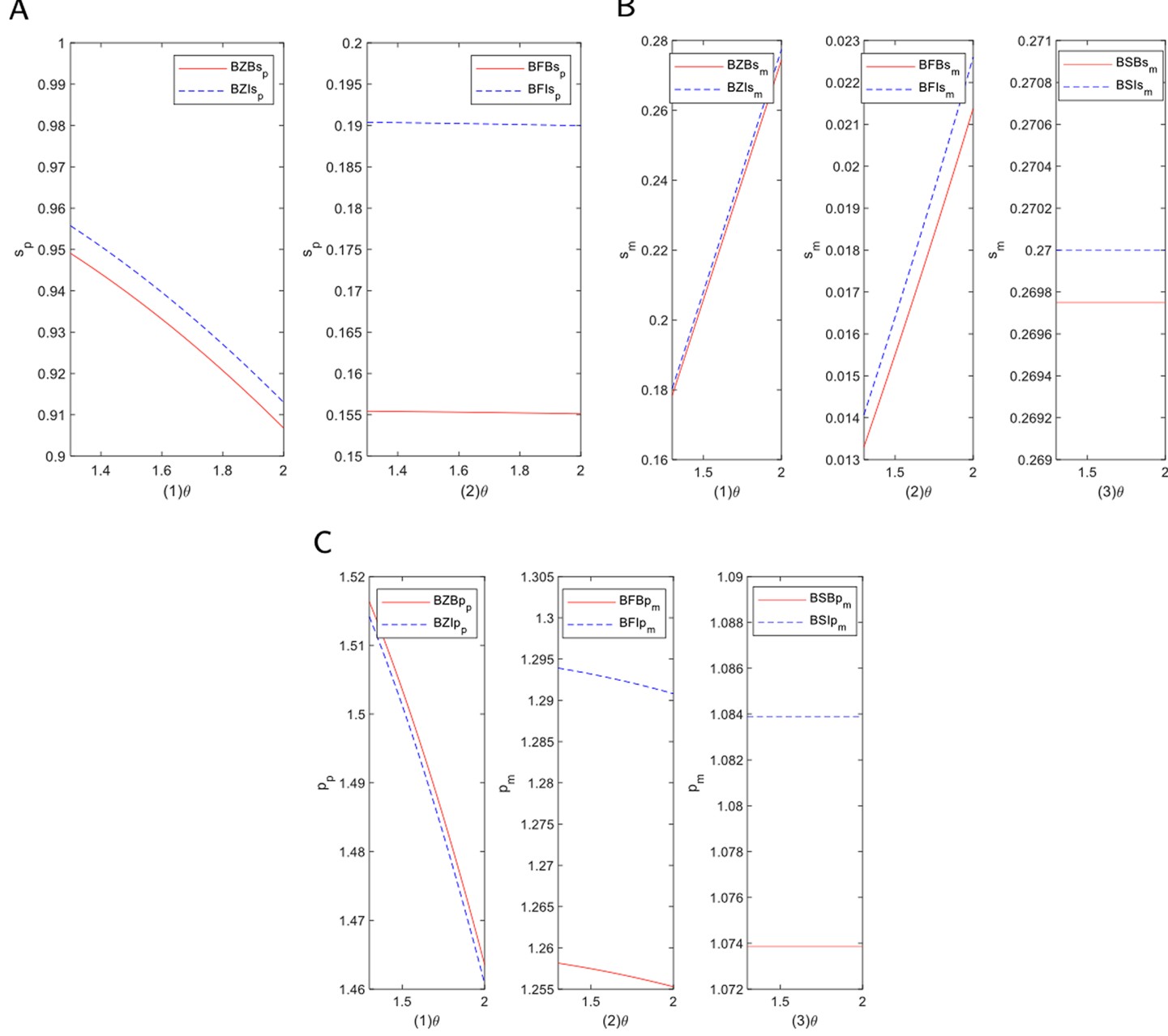

**Fig 5. Impact analysis of blockchain technology spillover effect.** (A) E-commerce service level. (B) Manufacturer service level. (C) Platform selling price.

entire operational process, growing demand for decentralization and transparency intensifies market competition. To control costs, the platform may reduce service investments, such as personalized services or logistics support, leading to a decline in service levels. This is particularly evident in external financing modes, where reducing operating costs is prioritized for short-term profit. In the FBP mode, outsourcing certain operations provides greater management flexibility. This model mitigates the service level decline by allowing the platform to focus on core competencies, thereby reducing direct

cost pressures. Consequently, the reduction in service level in the FBP mode is less pronounced than in the self-operated mode.

Fig 5B illustrates the dynamic trajectory of manufacturer service levels as the blockchain spillover effect is enhanced across the three modes. The service levels in the self-operated and FBP modes both show a significant upward trend. In sharp contrast, the curve for the SOP mode is nearly a horizontal line. The spillover effect, reflecting the growing influence of decentralization and transparency, impacts the relationships between manufacturers, platforms, and consumers. In the self-operated mode, the close relationship between the platform and manufacturer means that as blockchain improves supply chain efficiency and transparency, the platform demands a higher service level from manufacturers. Blockchain provides more accurate information, reducing uncertainties and enabling better quality control and process optimization by the manufacturer. In the FBP mode, blockchain enhances data sharing and collaboration between the platform and its third-party partners. This transparency improves supply chain responsiveness and allows manufacturers to better align with platform demands, thereby improving service levels. In contrast, the SOP mode's standardized approach and pre-set operating procedures limit its adaptability. Although blockchain can improve information flow, the impact on the manufacturer's service level is limited by the inherent rigidity of the established processes.

Fig 5C visually illustrates the price dynamics under the three modes as blockchain's influence grows. The curves for the self-operated and FBP modes both show a clear downward trend, while the price in the SOP mode remains almost horizontal. In the self-operated mode, where the platform controls procurement and distribution, blockchain technology enhances the efficiency of information flow and inventory management. This leads to a reduction in supply chain costs, such as those for inventory and transportation. As a result, the platform can lower sales prices to maintain price competitiveness. In the FBP mode, although the platform has less direct control over the supply chain, blockchain improves coordination with third-party partners. By ensuring more transparent and timely information transmission, blockchain reduces information asymmetry and operational friction, which in turn lowers overall supply chain costs and leads to a decrease in sales prices. Conversely, in the SOP mode, the platform adheres to standardized procedures and relatively fixed pricing systems. Although blockchain can improve transparency and efficiency, the existing cost structures and pricing strategies in this mode restrict significant price adjustments. Therefore, despite potential gains in operational efficiency, changes to the sales price in the SOP mode are minimal.

## 5.2 Decision on the introduction of blockchain technology

**5.2.1 Blockchain technology introduction in E-commerce platform.** To provide a clear decision-making framework for e-commerce platforms regarding the introduction of blockchain technology, we constructed a comprehensive profit analysis model. This model aims to reveal the ultimate impact of adopting blockchain on platform profits under various operating models, financing methods, and consumer preference levels. Fig 6 visualizes this complex decision-making landscape as a set of profit trend curves, where platform profit evolves with changes in consumer channel preference. First, in the self-operated mode, the introduction of blockchain has clear advantages. As shown in the figure, the profit curve representing "with blockchain" is consistently and significantly higher than the baseline "without blockchain" curve across the entire range of consumer preferences. This demonstrates that regardless of consumer preference, introducing blockchain is a beneficial decision for a self-operated platform. In this mode, blockchain allows the platform to optimize the supply chain, improve transaction transparency, and reduce fraud, thereby enhancing consumer trust and boosting sales efficiency. Consequently, the application of blockchain technology effectively drives profit growth for the platform, irrespective of the financing method.

Second, under the FBP and SOP modes, equity financing enables the platform to maximize the value derived from blockchain. Equity financing provides not only financial support but also additional resources to introduce new technologies, allowing the platform to gain a competitive advantage in technological innovation. In both modes, incorporating

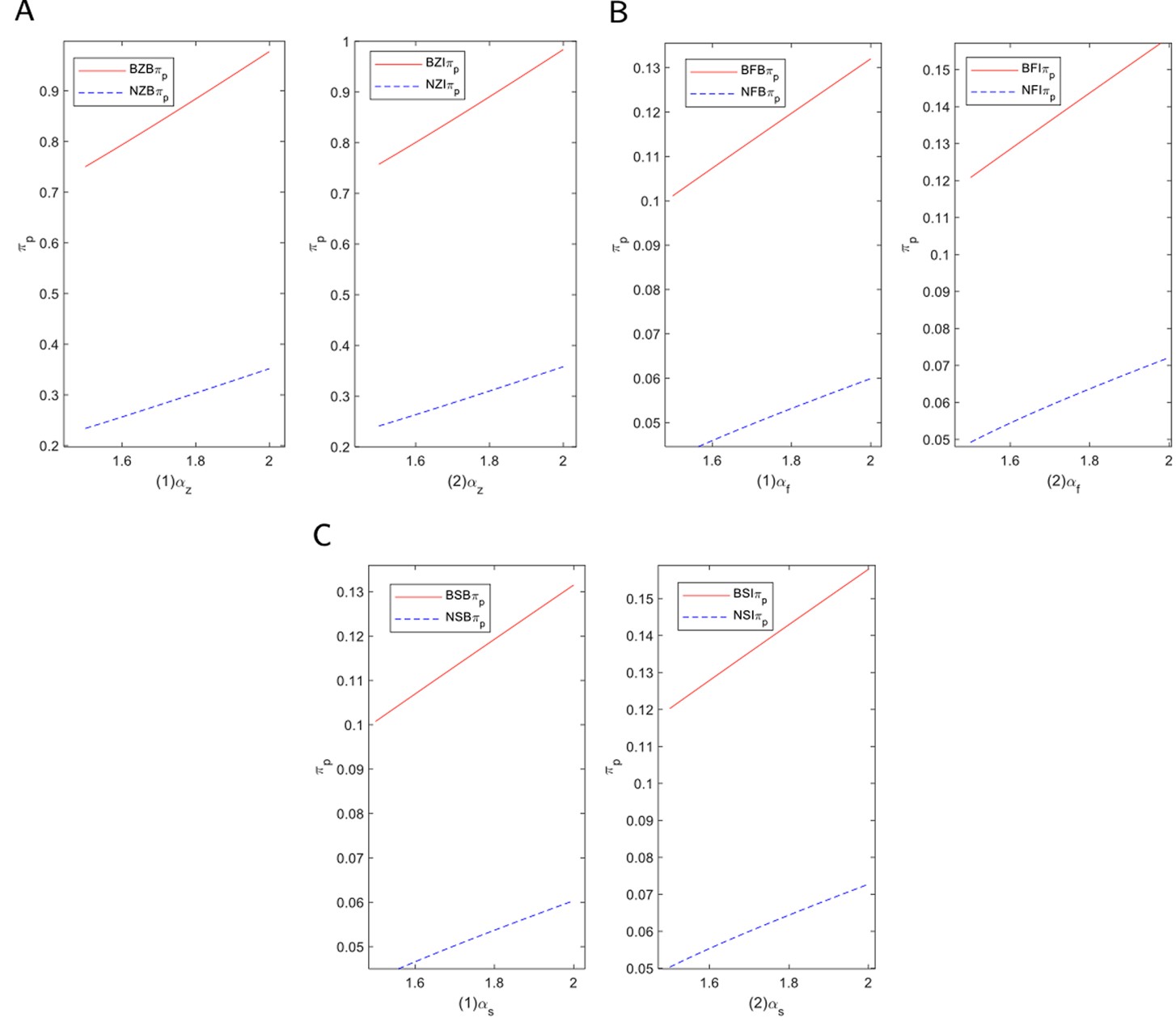

**Fig 6**. **Decision analysis of blockchain technology introduction in e-commerce platform.** (A) Self-operated mode. (B) FBP mode. (C) SOP mode.

blockchain technology helps the platform maintain a technological lead in a competitive market, thereby boosting its overall profitability.

However, when the platform opts for bank financing, consumer channel preference plays a crucial role in the decision-making process. Specifically, a threshold exists for consumer preference. If channel preference falls below this threshold, the e-commerce platform is more inclined to introduce blockchain technology to enhance consumer trust and the purchasing experience, thereby encouraging greater consumer engagement. On the other hand, if consumer channel preference

is high, it suggests that consumers already exhibit strong trust and loyalty toward the platform. In such cases, the platform may determine that introducing blockchain will not add significant value and may therefore choose to delay or forgo its application.

In summary, e-commerce platforms must consider different factors when deciding whether to introduce blockchain technology, depending on the financing mode and consumer preferences. For the self-operated mode, the introduction of blockchain generally enhances platform profits. In the FBP and SOP modes, equity financing allows the platform to better capitalize on the advantages of blockchain technology. In the case of bank financing, consumer channel preference becomes the key factor influencing the decision to apply blockchain technology.

**5.2.2 Blockchain technology introduction in manufacturer.** Following the platform-level analysis, we now turn to the supply chain's upstream to explore the profit outcomes for manufacturers participating in the implementation of blockchain technology. We examine how manufacturer profits evolve with consumer channel preferences to understand their decision-making rationale. Fig 7 visualizes these complex profit dynamics as a set of trend curves, where manufacturer profit changes with consumer channel preference. Firstly, in the proxy model, collaborating with the platform to implement blockchain is always the more profitable choice. As shown in the figure, the "with blockchain" profit curve is consistently and significantly higher than the "without blockchain" baseline curve across the entire range of preferences. This indicates that collaboration yields higher profits regardless of consumer preference levels. This is due to the close cooperation inherent in the proxy model, where the platform assumes greater responsibility for marketing and sales. The platform uses blockchain to optimize the supply chain and transaction processes, and manufacturers can easily integrate into this system, leveraging the technology to boost brand reputation and consumer trust, thus enhancing their market competitiveness.

Secondly, in the self-operated mode, the decision of whether to cooperate reveals a key "threshold effect." This is the most crucial finding in Fig 7A: the "with blockchain" profit curve intersects with the corresponding "without blockchain" baseline curve. When consumer channel preference is low, the "with blockchain" curve is positioned higher, indicating that cooperation brings additional profit. However, when consumer channel preference is high, the "with blockchain" curve crosses below the baseline, indicating that cooperation at this stage would lead to a profit loss. This intersection represents the critical point at which the manufacturer's optimal decision reverses. This is because, in the self-operated mode, manufacturers have less control over the e-commerce platform and must make independent decisions, which entails greater risk and encourages more caution when adopting blockchain. To maximize benefits, they typically conduct a more detailed evaluation of the technology's implementation costs—including expenses for hardware, software development, and training—and weigh them against the potential advantages blockchain can bring, such as enhanced consumer trust and improved product traceability.

In this mode, the manufacturer's decision is largely driven by a cost-benefit analysis, rather than by simply following technological trends or responding to external pressures. By carefully assessing the value blockchain brings to consumers, manufacturers can determine whether the benefits justify the costs of adoption. If the technology's costs are too high or if the return on investment is not clear in the short term, manufacturers may opt to delay or forgo its application. As a result, in the self-operated mode, manufacturers are more focused on tangible economic returns and long-term benefits from blockchain technology, rather than simply on embracing technological innovation or market trends.

## 5.3 Financing decisions under blockchain technology

To determine how manufacturers should choose an optimal financing strategy after introducing blockchain, we constructed a financing decision domain model. This model provides a quantitative framework for choosing between options such as bank loans and equity financing. Fig 8 visualizes this framework as a decision region map, where the horizontal axis represents unit production cost and the vertical axis represents the direct channel product price. Together, these two core indicators determine the enterprise's profit margin and financial risk. The coordinate plane is accordingly divided into

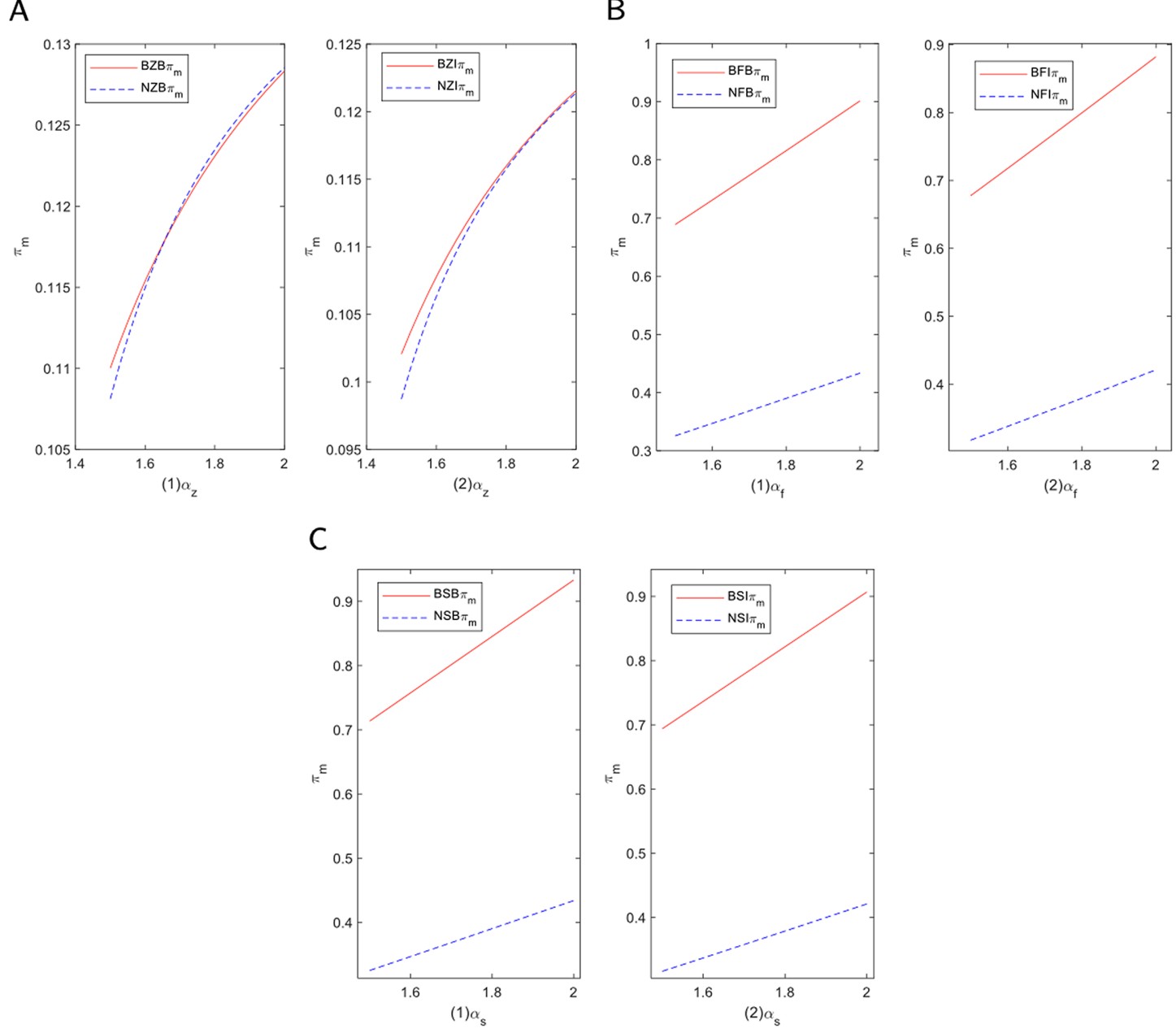

**Fig 7. Decision analysis of blockchain technology introduction in manufacturer.** (A) Self-operated mode. (B) FBP mode. (C) SOP mode.

distinct financing decision domains: Region I represents the optimal zone for bank financing, while Region IV represents the optimal zone for equity financing. The regional division in this figure reveals key patterns. First, the decision domain for bank financing (Region I) is located in the upper-left corner, corresponding to a combination of low production costs and high product prices. This indicates that bank loans are the preferred financing method for enterprises with high profit margins and healthy financial conditions. In sharp contrast, the decision domain for equity financing (Region IV) occupies the bottom-right corner, representing a scenario of high production costs and low product prices. This reveals that when

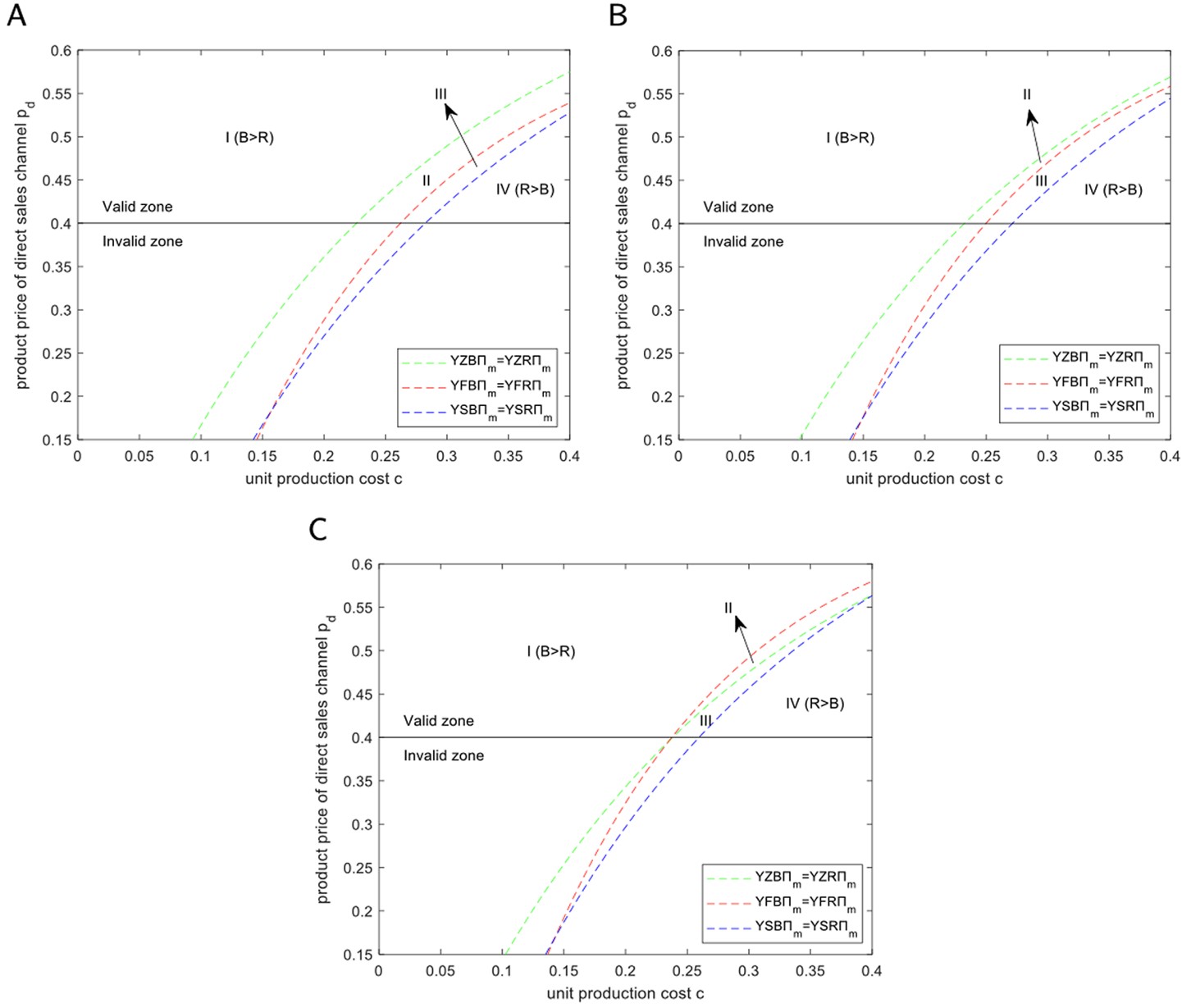

**Fig 8. Analysis of manufacturer's blockchain technology financing decision.** (A) $c_b = 0.02$. (B) $c_b = 0.05$. (C) $c_b = 0.08$.

a company's profit margin is severely squeezed and operating risks are high, equity financing becomes the more viable financing channel.

The preference for bank financing in the "safe zone" (Region I) stems from the risk-averse nature of banks. Enterprises in Region I have high profit margins that ensure strong cash flow and debt repayment ability, making them high-quality borrowers for banks. In this situation, the application of blockchain technology further enhances their advantages. By improving supply chain and capital flow transparency, blockchain acts as a powerful credit enhancement tool, reducing banks' audit costs and credit risks and enabling manufacturers to obtain loans on more favorable terms.

Equity financing becomes the solution in the "challenge zone" (Region IV) because the high-risk profile of these enterprises makes it difficult for them to obtain bank loans. When production costs are high and market prices are low, the enterprise's profitability is weak and default risk is high, leading banks to typically refuse loans. At this point, companies must turn to equity financing, transferring partial ownership in exchange for necessary funds. Investors value a company's future growth potential more than its current profitability, which is a fundamentally different logic from that of banks.

In summary, Fig 8 indicates that the optimal financing strategy is not a subjective choice but is objectively determined by the enterprise's core business indicators (cost and price). It also demonstrates the dual value of blockchain: for healthy enterprises, it acts as a "booster" for obtaining low-cost bank loans; for enterprises facing challenges, the transparent system it builds serves as the "cornerstone of trust" to attract equity investors.

## 6 Case studies

Zhizhen Chain is a technology sub-brand of JD Technology (formerly JD Digits) that focuses on providing specialized blockchain solutions [47]. Since 2016, JD Group has invested in blockchain technology R&D, exploring its applications across various sectors. The company's first large-scale application, a product anti-counterfeiting and traceability system, initiated the widespread commercial use of blockchain within JD and has since achieved significant scale. This application has expanded from a single product category to encompass multiple categories. JD's e-commerce model, which leverages an integrated warehousing and distribution system, facilitates close cooperation with upstream suppliers. This operational structure provides a distinct advantage for the development of "blockchain + anti-counterfeiting traceability" solutions, enabling efficient, accurate, and transparent commercial operations that contribute to JD's long-term growth and competitive edge.

As a leading blockchain-enabled platform in China, JD's Zhizhen Chain provides technical support across supply chain segments through three modes: Self-operated, FBP, and SOP. This section analyzes the impact of blockchain technology on supply chain decisions through practical cases. By combining financial and operational data, we explore how blockchain influences service, pricing, and financing decisions. The analysis examines representative brand cases under each of JD Zhizhen Chain's three modes: the Self-operated mode, represented by JD Health's pharmaceutical traceability system for brands like Tongrentang; the FBP mode, represented by JD Logistics' smart home appliance service for Midea; and the SOP mode, represented by Chow Tai Fook's gold traceability system.

### 6.1 JD health case

JD Health has successfully established an end-to-end pharmaceutical traceability system using its "Zhizhen Chain" blockchain, achieving significant advancements in several areas. The core of this system is the use of blockchain technology to provide information transparency and traceability, enabling the real-time monitoring and verification of each step in the pharmaceutical supply chain. This implementation has enhanced service efficiency, strengthened pricing power, and reduced financing costs. A detailed analysis of Zhizhen Chain's impact on JD Health across these three domains is provided below.

1. Enhanced Service Efficiency
By leveraging the decentralized nature of blockchain technology, JD Health has significantly improved the transparency and efficiency of its service processes. By minimizing intermediaries in information transmission, the company notably reduced its after-sales service response time, which in turn lowered the service cost coefficient. Specifically, the after-sales response time decreased from 72 hours to 30 hours, a reduction of 58% (Blockchain Pharmaceutical Application White Paper, 2022). This outcome demonstrates blockchain's advantage in mitigating information asymmetry while enhancing information sharing and automated processing. The technology enables JD Health to identify and respond to customer issues more efficiently, reducing costs associated with information delays or

errors and resulting in a smoother and more accurate service system. This improved service efficiency increases both customer satisfaction and overall operational effectiveness.

2. Strengthened Pricing Power

The introduction of blockchain technology has empowered JD Health with stronger pricing power for its pharmaceutical products. Products certified by "Zhizhen Chain" command prices that are, on average, 12.3% higher than their uncertified equivalents. This premium stems from the enhanced trust conferred by blockchain, as consumers exhibit greater confidence in certified medicines and are willing to pay more. This validates blockchain's potential to increase the perceived value of a product, particularly in the healthcare sector where supply chain transparency builds consumer trust and can justify higher pricing. Furthermore, the implementation of blockchain effectively reduced price elasticity; after price adjustments, the corresponding drop in demand was only 5.1% (China Pharmaceutical E-commerce Consumer Insights Report, 2023). This indicates that by boosting market trust, blockchain grants JD Health greater negotiating leverage and flexibility in its pricing strategies, creating larger profit margins and strengthening its competitive market position.

3. Reduced Financing Costs

JD Health has also achieved significant improvements in financing through blockchain technology. By pledging its on-chain data assets as collateral, JD Health secured lower bank loan interest rates, reducing them by 1.8 percentage points compared to traditional financing modes (Fintech-Enabled Supply Chain Finance Practice Case Collection, 2023). This not only lowered the company's financing costs but also demonstrated blockchain's broad application potential in finance. The technology enables banks and financial institutions to assess JD Health's credit risk more accurately, as all transaction and asset data are traceable on-chain. This reduces information asymmetry and makes lending risks more manageable.

## 6.2 Midea case

Through the FBP mode, JD Logistics provided Midea with blockchain-enabled warehousing and distribution services, yielding significant benefits across multiple domains. This innovative approach optimized supply chain management and led to notable improvements in service quality, inventory control, and equity financing. The specific effects of this blockchain implementation are analyzed below:

1. Service Spillover Effect

The application of blockchain technology to Midea's warehousing and distribution not only improved online service quality but also significantly reduced offline channel return rates. By enhancing information sharing and transparency, blockchain streamlined the returns process, which increased consumer trust and consequently reduced the frequency of returns. Specifically, Midea's offline channel return rate decreased by 17% (Retail Supply Chain Service Quality Index, 2023). This result demonstrates a cross-channel benefit, as data sharing enabled synergistic optimization and an improved overall service experience. Furthermore, the decentralized and immutable recording of transactions boosted consumer confidence at the point of purchase, while the accurate traceability of return reasons minimized disputes and improved customer relationship management.

2. Platform-Vendor Decision Synergy

The use of blockchain in demand forecasting significantly improved Midea's inventory management. By leveraging real-time on-chain data and smart contracts, Midea could more accurately predict market demand, leading to optimized production planning and inventory scheduling. This resulted in a 32% increase in inventory turnover (Empirical Research on Manufacturing Inventory Optimization), which effectively reduced inventory backlogs, lowered warehousing costs, and improved capital efficiency. This capability also enhanced decision-making synergy between the platform and the vendor. Real-time data sharing via blockchain ensured tighter coordination between JD Logistics and Midea, allowing for closer alignment of production and logistics services. By basing decisions on a single,

shared data source, both parties could avoid biases from information asymmetry and achieve more efficient supply chain management.

3. Equity Financing Preference

To jointly fund the blockchain deployment, JD and Midea established an innovation fund to support the technology's further application and optimization. This initiative alleviated Midea's direct investment burden and allowed it to improve liquidity through equity financing. Under this model, Midea's profits increased by 21% (Industrial Blockchain Economic Evaluation Model), demonstrating the advantages of using equity financing in blockchain collaborations. The joint innovation fund not only fostered deeper cooperation between Midea and JD Logistics but also provided a successful financing framework for other manufacturers. By using equity financing, Midea secured necessary funding without incurring significant debt pressure, which in turn drove further blockchain innovation. This practice showcases the potential of combining blockchain technology with equity financing under the FBP mode to create mutually beneficial outcomes.

### 6.3 Chow Tai Fook case

Under the SOP mode, Chow Tai Fook independently deployed blockchain technology for its gold traceability system, making significant advancements that enhanced its competitiveness and market performance. By leveraging blockchain's transparency and traceability, Chow Tai Fook secured notable advantages in cost, financing, and channel competition, providing an innovative model for future operations. The specific outcomes of this blockchain implementation are analyzed below:

1. Cost-Effective Decisions

In the SOP mode, Chow Tai Fook bore the full cost of its blockchain deployment, a strategy that directly contributed to lower service costs compared to traditional self-operated modes. The latter typically incurs additional expenses related to intermediaries, inventory management, and multi-stage supply chain operations. Blockchain technology significantly reduced overall service costs by streamlining processes, enhancing information transparency, and lowering intermediary fees. This mode allowed Chow Tai Fook to provide high-quality service at a lower cost, aligning with market demand for both efficiency and affordability. This phenomenon is consistent with the theory that service investment tends to decrease under high-cost conditions. Blockchain enabled Chow Tai Fook to automate its supply chain and improve transparency, which reduced labor, material consumption, and other operational costs.

2. Innovative Financing Methods

Beyond cost advantages, blockchain technology enabled financing innovation for Chow Tai Fook. Using its on-chain transaction data, the company successfully issued Asset-Backed Securities (ABS) for financing. This method, which used gold transaction data as the underlying assets for the securities, carried an interest rate of 4.2%, which was 2.3 percentage points lower than that of traditional bank loans (Shanghai Stock Exchange, 2023 Asset Securitization Market Report). This innovative approach not only reduced financing costs but also expanded financing channels and improved liquidity. This case supports the conclusion that in high-cost scenarios, equity financing offers advantages over traditional bank loans. The transparency and credibility of on-chain transaction data allow investors to better assess asset value, thereby reducing financing risk and costs. Through securitization, Chow Tai Fook improved its capital utilization efficiency and demonstrated a replicable financing mode.

3. Reduced Channel Competition

Historically, price differences between online and offline channels have significantly influenced consumer purchasing decisions, particularly in the gold jewelry industry. Chow Tai Fook successfully narrowed this price gap using its gold traceability blockchain, reducing it from a traditional 22% to just 8% (World Gold Council, 2024 China Gold Jewellery Consumption Trends Report). This reduction reflects blockchain's effectiveness in mitigating channel conflict

and enhancing price consistency. By providing consumers with end-to-end traceability data via the on-chain system, Chow Tai Fook ensured consistent information and pricing across all channels. This transparency alleviated consumer concerns about price discrepancies, thereby boosting brand trust. Simultaneously, this approach diminished the competitive friction between online and offline channels, creating a fairer and more transparent purchasing environment for consumers.

This chapter has explored the practical application of blockchain technology in supply chain decision-making through a detailed analysis of representative partner brands under JD Zhizhen Chain's three modes. By providing quantified analysis based on these real-world cases, it validates the impact of blockchain on service, pricing, and financing.

## 7 Management insights

This study provides practical management insights for key stakeholders within the supply chain finance ecosystem, including e-commerce platform managers, manufacturers (specifically small and medium-sized enterprises), and policy-makers. These insights are intended to guide these parties in making more informed and strategic decisions regarding the introduction of blockchain technology, the selection of financing methods, and the choice of operating modes.

### 7.1 Management insights for E-commerce platforms

The operational mode of an e-commerce platform is central to its strategic decision-making regarding blockchain adoption. This research indicates that the self-operated mode yields the highest return on investment in blockchain technology, offering stronger advantages in service levels and pricing power. Therefore, platform managers should prioritize the integration of blockchain technology within their self-operated businesses to directly increase profits and strengthen competitive barriers.

For the FBP and SOP modes, decision-making requires a more nuanced approach. When manufacturers on the platform utilize equity financing, the introduction of blockchain can yield substantial benefits, and the platform should actively encourage its adoption. Conversely, if manufacturers rely on bank financing, the platform must carefully assess consumer channel preferences. When consumer preference for offline channels falls below a specific threshold, introducing blockchain becomes a worthwhile investment, as it can build trust in the online channel and attract more customers. To this end, platform managers should develop robust market insight mechanisms to continuously monitor shifts in channel preferences, enabling them to identify the optimal timing for blockchain implementation.

### 7.2 Management insights for small and medium sized enterprises (manufacturers)

For manufacturing enterprises within the supply chain, financing decisions are critical for survival and growth. This study suggests that a manufacturer's financing choices should be closely linked to its cost structure. When production costs are low, prioritizing lower-cost bank financing is advisable. Conversely, when production costs are high and offline sales prices are under pressure, equity financing becomes the more prudent choice to diversify risk and secure resources.

The introduction of blockchain technology is a key variable that alters this landscape. Manufacturers must recognize that implementing blockchain can shift their optimal financing strategy. As the cost of blockchain application increases, bank financing becomes more attractive under the self-operated mode, whereas equity financing is the preferred option under the FBP and SOP modes. Therefore, when negotiating blockchain adoption with platform partners, manufacturers must consider the long-term implications for financing costs and flexibility. They should select the financing combination that best aligns with their financial position to ensure sustainable operations.

### 7.3 Management insights for policy makers

As guides and regulators of the digital economy, policymakers play a pivotal role in promoting innovation in supply chain finance. The findings of this study indicate that the benefits of blockchain technology are contingent upon the business model in which it is applied. Therefore, policy support should avoid a one-size-fits-all approach and instead be differentiated.

First, policymakers should encourage and potentially subsidize the piloting and promotion of blockchain technology within self-operated e-commerce modes, as this is where its social benefits and spillover effects are most pronounced. Second, to address the high cost of blockchain adoption for small and medium-sized enterprises under the FBP and SOP modes, policymakers could establish data standards, support the development of public infrastructure, or offer tax incentives to lower the technological barriers to entry. Finally, it is essential to enhance the regulatory framework and information disclosure requirements for equity financing linked to blockchain applications. This framework should aim to encourage financial innovation and expand financing channels for SMEs while simultaneously protecting investor rights and preventing systemic risk, thereby fostering a healthy, transparent, and trustworthy supply chain finance ecosystem.

## 8 Conclusion with future studies

### 8.1 Conclusion

This paper integrates blockchain technology, different e-commerce platform entry modes, and dual-channel sales into a supply chain finance framework. It examines the factors influencing manufacturer and e-commerce platform profits under two distinct financing methods: bank financing and equity financing. By comparing scenarios with and without blockchain, the paper analyzes the technology's impact on channel preferences, its spillover effects on supply chain decisions, and the optimal conditions for its adoption alongside corresponding financing strategies. Through theoretical modeling and numerical analysis, the following conclusions are drawn:

1. In the self-operated mode, decision variables exhibit greater sensitivity to the parameters associated with blockchain implementation. In contrast, the magnitude of change in these variables is more moderate in the FBP and SOP modes. Furthermore, the online service level, offline service level, and online sales price are consistently higher in the self-operated mode than in the other two modes. This indicates that the self-operated mode offers inherent advantages in service quality and pricing flexibility, thereby enhancing its overall competitiveness.

2. The decision to adopt blockchain technology is contingent not only on the financing method but also on channel preferences and the platform's business model. In the self-operated mode, the platform is always incentivized to introduce blockchain, as it increases profitability regardless of the manufacturer's chosen financing method. In the FBP and SOP modes, when a manufacturer opts for equity financing, the platform also gains significant benefits, making adoption the favorable choice. However, if the manufacturer uses bank financing in these modes, the platform's decision depends on channel preferences. Specifically, if consumer preference for the offline channel falls below a certain threshold, the platform will be inclined to introduce blockchain to enhance online consumer trust, thereby boosting its competitiveness.

3. Regardless of the entry mode, manufacturers with low production costs tend to prefer bank financing after blockchain implementation, while those with high production costs and low offline sales prices favor equity financing. As the application cost of blockchain increases, the optimal decision domain for bank financing expands under the self-operated mode, while the domain for equity financing expands under the FBP and SOP modes.

## 8.2 Limitations

While this study provides a detailed exploration of the interplay between blockchain technology, financing methods, and e-commerce models, its research framework and underlying assumptions introduce certain limitations. These limitations, in turn, highlight valuable directions for future research.

This study has the following limitations. First, the model is constructed upon several idealized assumptions, including a linear market demand function and a homogeneous consumer valuation of the benefits provided by blockchain. Real-world consumer behavior is often more complex, and preferences for blockchain-enabled transparency may exhibit non-linear or heterogeneous characteristics, potentially limiting the generalizability of the findings. Second, the analysis focuses exclusively on bank financing and equity financing. This scope does not encompass the increasingly diverse range of supply chain finance instruments, such as trade credit and asset-backed securities (ABS). The exclusion of these alternatives from the unified analytical framework is another limitation of this research.

## 8.3 Future research

Building upon the limitations identified, future research can be extended along the following dimensions. First, future studies could incorporate more sophisticated models of consumer behavior, such as utility theory or prospect theory, to better characterize the trust premium and purchasing decisions related to blockchain technology. This approach would facilitate the development of demand functions that more accurately reflect real-world market dynamics.

Second, the scope of financing methods could be expanded to include other innovative financial instruments, such as online crowdfunding and supply chain ABS. An analysis comparing the applicability and efficiency of these tools within a blockchain-enabled environment would provide a more comprehensive financing framework for small and medium-sized enterprises.

Finally, the static, single-period nature of the current model could be addressed by developing a multi-stage dynamic game model. Such a framework would allow for an analysis of the long-term evolution of blockchain's spillover effects and would enable an examination of how enterprises might dynamically adjust their blockchain adoption and financing strategies across different developmental stages.

## Supporting information

**S1 Data. Simulation parameters are set in the file S1 Data.**
(DOCX)

**S1 Appendix. All detailed proof processes are included in the document S1 Appendix.**
(TEX)

## Acknowledgments

We are very grateful to the attention to detail of our editor and three reviewers, whose comments have markedly improved the manuscript.

## Author contributions

**Methodology:** Shangyu Pi.

**Visualization:** Shangyu Pi.

**Writing – original draft:** Shangyu Pi.

**Writing – review & editing:** Shangyu Pi.

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
