## [Decision Letter · Decision Letter 0]

20 Aug 2025

PONE-D-25-31648The Impact of Blockchain Adoption on Supply Chain Financing and E-Commerce Platform DynamicsPLOS ONE

Dear Author,

Thank you for submitting your manuscript to PLOS ONE. After careful consideration, we feel that it has merit but does not fully meet PLOS ONE’s publication criteria as it currently stands. Therefore, we invite you to submit a revised version of the manuscript that addresses the points raised during the review process.

We look forward to receiving your revised manuscript.

Kind regards,

Chia-Huei Wu

Academic Editor

PLOS ONE

Journal Requirements:

 A clean copy of the edited manuscript (uploaded as the new *manuscript* file).

“1. Institutions Young Innovative Talent Program�2023WQNCX039), Sponsor: Department of Education of Guangdong Province;

2. 2021 Guangdong Polytechnic Normal University Talents Research Project, Sponsor: Guangdong Polytechnic Normal University. “

“This research was funded by 1. Institutions Young Innovative Talent Program�2023WQNCX039), Sponsor: Department of Education of Guangdong Province; 2. 2021 Guangdong Polytechnic Normal University Talents Research Project, Sponsor: Guangdong Polytechnic Normal University.”

“1. Institutions Young Innovative Talent Program�2023WQNCX039), Sponsor: Department of Education of Guangdong Province;

2. 2021 Guangdong Polytechnic Normal University Talents Research Project, Sponsor: Guangdong Polytechnic Normal University.”

Additional Editor Comments (if provided):

Major Revision

Reviewers' comments:

Reviewer's Responses to Questions

**Comments to the Author**

1. Is the manuscript technically sound, and do the data support the conclusions?

Reviewer #1: Partly

Reviewer #2: No

Reviewer #3: Partly

2. Has the statistical analysis been performed appropriately and rigorously?

Reviewer #1: I Don't Know

Reviewer #2: No

Reviewer #3: No

3. Have the authors made all data underlying the findings in their manuscript fully available?

Reviewer #1: No

Reviewer #2: No

Reviewer #3: Yes

4. Is the manuscript presented in an intelligible fashion and written in standard English?

Reviewer #1: No

Reviewer #2: Yes

Reviewer #3: Yes

5. Review Comments to the Author

Reviewer #1: Dear Editor,

Thank you for inviting me to review this manuscript. The study addresses an important and timely topic by examining blockchain adoption in dual-channel supply chains, integrating financing strategies and platform dynamics. While the research shows promise, I recommend major revisions to strengthen the manuscript before publication. Below are my specific comments:

Major Concerns

1. The manuscript has many grammatical issues. It would benefit from thorough professional English editing.

2. Literature Review lacks critical synthesis. Many citations are presented in list format without clearly identifying the research gap.

3. The realism of these assumptions and possible extensions in the limitations or discussion section should be further discussed.

4. The mathematical results are comprehensive, but no empirical data or simulation is used to validate the model or demonstrate sensitivity.

5. Figures/tables (e.g., Figure 4) are referenced but not well-integrated into the text. Their insights should be explained more clearly.

6. Highly technical appendix proofs could be summarized in the main text or moved to supplementary materials.

7. Recommendation: Include numerical simulations or illustrative graphs showing how key parameters (e.g., service sensitivity γ, blockchain cost cb, financing type) affect decisions and profits.

8. The article lacks a dedicated section translating results into actionable insights for practitioners (e.g., when platforms should offer financing or adopt blockchain).

Regards

Reviewer #2: I am really grateful to review this manuscript. In my opinion, this manuscript can be published once some revision is done successfully. I made one suggestion and I would like to ask your kind understanding.

This study develops a two-level supply chain model consisting of a financially constrained manufacturer (M) and an e-commerce platform (P) that utilizes blockchain technology. The study examines the impact of platform channel preferences and service levels across three merchant entry modes: the platform self-operated mode, Fulfillment by Platform (FBP) mode, and Sales on Platform (SOP) mode. A game-theoretic model, based on Stackelberg theory, is constructed to integrate both bank financing and equity financing in analyzing production and service decisions following blockchain implementation. This study explores the decision-making processes of both the e-commerce platform and the manufacturer regarding blockchain adoption and their respective financing strategies. Based on the results of this study, the adoption of blockchain increases profits for both manufacturers and platforms; however, decision variables exhibit different sensitivities depending on the entry mode. The self-operated mode offers more flexibility in service levels and pricing, while the FBP and SOP modes tend to show more stable decision dynamics. Moreover, the decision to implement blockchain is influenced by financing methods and channel preferences, with e-commerce platforms being more inclined to adopt blockchain in the self-operated mode, particularly under equity financing. Manufacturers also exhibit distinct financing preferences based on production costs, with bank financing being more advantageous at lower production costs and equity financing preferred when costs are higher. These findings provide valuable insights into how blockchain technology can impact supply chain dynamics, financing decisions, and the overall competitiveness of platform-based business models. I would argue that this is a good start.

However, it can be noted that every simulation study including this study has an issue of external validation, i.e., how to validate models with external data. In this context, I would like to ask the authors to address this issue in great detail.

Reviewer #3: Reviewer’s Comments to the Editor and Authors

Manuscript ID: PONE-D-25-31648

Title: The Impact of Blockchain Adoption on Supply Chain Financing and E-Commerce Platform Dynamics

Recommendation: Major Revision

General Assessment

This manuscript takes a deep dive into how adopting blockchain technology affects service levels, financing options, and pricing strategies in dual-channel e-commerce supply chains, all through a detailed Stackelberg game-theoretic model. The research question is not only timely but also highly relevant, and the mathematical framework used is quite rigorous. The different scenarios explored—platform self-operated, Fulfillment by Platform (FBP), and Sales on Platform (SOP)—are well-defined, and the findings could be valuable for both academics and managers alike.

That said, the manuscript needs quite a bit of work to improve clarity, update the literature review, and better link the model to real-world implications. Though the complexity of the mathematical derivations is a strong point, it also makes the content less accessible. Additionally, there are no robustness checks, a lack of integration with the latest related studies, and no clear statement of limitations. Overall, the manuscript is understandable, but the English could use some refinement to enhance flow, conciseness, and readability. The main issues include lengthy sentences, occasional awkward phrasing, and inconsistent technical terms. Although these issues may not fully obfuscate the idea, they do reduce clarity and professionalism in the presentation.

MAJOR ISSUES

1.Clarity and Accessibility

The presentation is quite heavy on the math, which might make it tough for the wider PLOS ONE audience to grasp. Consider summarizing the key findings in tables and diagrams to help clarify the comparative results across different scenarios. A conceptual flow diagram illustrating the decision-making process would also be beneficial.

2.Literature Review

While the literature review does a good job in some areas, it misses out on recent research related to blockchain-enabled supply chains from 2022 to 2024. Refresh the review to include the latest empirical and theoretical studies, particularly those focusing on blockchain adoption frameworks, cost-benefit analyses, and real-world validations in supply chain settings.

3.Model Assumptions

Some assumptions (like the uniform distribution of consumer valuations, fixed commission rates, and external blockchain costs) need a bit more backing. Offer empirical or literature-based justifications for these assumptions, and discuss how relaxing them could impact the results.

4.Comparative and Sensitivity Analysis

The comparative analysis currently only looks at closed-form solutions. It doesn’t explore how sensitive the results are to key parameters like blockchain cost , consumer service sensitivity (γ), and spillover effects (θ). Add sensitivity analyses or numerical simulations to evaluate the robustness of the conclusions.

5.Managerial and Policy Implications

Although there are some managerial insights provided, they tend to be brief and mostly qualitative. Broaden this section with more specific, actionable implications for platform managers, SMEs, and policymakers, drawing from the model results.

6.Limitations and Future Research

The current conclusion doesn’t clearly address limitations or future research opportunities. Include a dedicated subsection that discusses the model's scope and constraints, and suggest potential directions for further empirical validation and model enhancements.

7.Sentence Length and Structure

A lot of sentences are quite lengthy and packed with multiple clauses, making them tricky to follow, especially in the Introduction and Literature Review. For instance, lines 32–45 in the Introduction throw together statistics, examples, and conceptual discussions into one long-winded sentence. This can lead to reader fatigue and makes it tough to grasp the arguments. Try breaking down those complex sentences into shorter ones, each zeroing in on a single idea.

8.Terminology Consistency

Terms like “self-operated mode,” “proprietary mode,” and “platform self-operated mode” are tossed around interchangeably, as are “FBP mode” and “Fulfillment by Platform.” Additionally, mathematical symbols (like , , ) sometimes pop up without a reminder of their meanings in later sections. This can confuse readers who aren’t familiar with the terminology. Standardize the terms and create a consistent glossary early on.

MINOR ISSUES

I.Abstract: Simplify the language for readers who aren’t specialists; also, briefly mention any limitations.

II.Notation Consistency: Make sure all variables are clearly defined before they’re used; some symbols in the Appendices show up without any explanation.

III.Proof Presentation: In Appendices A–C, consider adding short explanations between steps to help readers follow along more easily.

IV.Word Choice and Redundancy

There are moments where the same idea is repeated with slightly different wording, like when “combining blockchain and different entry modes” pops up multiple times in the Introduction. Plus, phrases like “at present,” “therefore,” and “thus” are used a bit too often without really adding clarity. This makes the text feel a bit bloated. Cut out the redundancies and tighten up the wording.

V.Grammar and Agreement

There are a few subject-verb agreement errors and some issues with article usage, such as “the manufacturer determines , the service level” where there's a missing article or spacing. While it doesn’t majorly affect understanding, it does detract from the overall polish. A careful proofreading or professional language editing would help.

VI.Flow and Transitions

The transitions between subtopics in the literature review can feel a bit jarring, leaving readers to piece together the logical connections on their own. This reduces the overall narrative cohesion. Adding brief linking sentences can help guide readers smoothly from one theme to the next.

VII.Use of Passive Voice

There’s a noticeable reliance on passive voice in the Methods and Literature Review sections (“It is assumed that…”, “It is found that…”) which can lessen engagement. While not incorrect, it can make the writing feel less direct. Opt for active voice where it makes sense, especially when discussing the authors’ own contributions.

GENERAL SUMMARY

The manuscript has a lot of potential, but it could really use some revisions to align with PLOS ONE’s standards for clarity, relevance, and practical application. By addressing the earlier points, the authors can significantly boost both the scholarly and practical impact of their work. While the overall clarity is decent, there are a few highlighted issues—particularly with sentence length and inconsistent terminology—that need attention. It will require moderate to major editing to tackle these structural and terminology challenges. The authors might want to think about reaching out to professional academic English editing services before they resubmit.

RECOMMENDATION: MAJOR REVISION

6. PLOS authors have the option to publish the peer review history of their article (what does this mean?). If published, this will include your full peer review and any attached files.

Reviewer #1: No

Reviewer #2: No

Reviewer #3: No

---

## [Author Response · Author response to Decision Letter 1]

9 Sep 2025

Dear Editor,

Thank you for the opportunity to revise our manuscript titled “The Impact of Blockchain Adoption on Supply Chain Financing and E-Commerce Platform Dynamics” (ID: PONE-D-25-31648). We are very grateful to you and the anonymous reviewers for the time and effort invested in providing such thorough and constructive feedback. The comments were invaluable and have guided us in significantly improving the quality, clarity, and contribution of our paper.

We have carefully considered every comment and have revised the manuscript accordingly. Major revisions include: strengthening the introduction and literature review, adding a dedicated section for model assumptions and a table of notations, moving detailed proofs to supplementary materials while adding proof sketches, and most substantially, completely overhauling the numerical analysis section and adding an entirely new section on actionable management insights.

Below, we provide a point-by-point response to the comments from Reviewers. For ease of reference, we have highlighted all major changes in the revised manuscript.

Reviewer #1:

Major Concerns

1. The manuscript has many grammatical issues. It would benefit from thorough professional English editing.

Response 1: We are very grateful for this valuable suggestion. We agree that the quality of language is critical, and we apologize that the previous version did not meet the required standard. In response to your comment, the manuscript has undergone a thorough professional English editing process. Great efforts have been made to improve the language, correcting all grammatical and stylistic issues to ensure clarity and precision. We hope the revised manuscript is now suitable for publication.

2. Literature Review lacks critical synthesis. Many citations are presented in list format without clearly identifying the research gap.

Response 2: We are very grateful for this constructive and insightful comment. We agree that the original literature review was more descriptive than synthetic and did not adequately highlight the research gap. Following your suggestion, we have substantially restructured and rewritten the entire Literature Review section (Section 2). Specifically, we have moved away from a ‘list format’ and have now organized the review around key themes and debates to provide a more critical synthesis. Crucially, and in direct response to your comment, we have added a concluding summary at the end of each subsection to clearly articulate the specific research gap. Furthermore, to enhance clarity, we have included a new table (Table 1) to chronologically position our work within the field. The revised section can be found on pages 4-7 of the updated manuscript (Please refer to the red text on lines 117-234 on pages 4-7). We believe these extensive revisions now provide a strong, logical foundation for our study.

3. The realism of these assumptions and possible extensions in the limitations or discussion section should be further discussed.

Response 3: We thank the reviewer for this insightful suggestion. We agree that a discussion of our model’s assumptions and potential extensions is crucial. Accordingly, we have added two new dedicated subsections to the Conclusion: Section 8.2 ‘Limitations’ and Section 8.3 ‘Future Research’ (Please refer to the red text on lines 1179-1211 on page 34). In the ‘Limitations’ subsection, we now discuss the realism of core assumptions, such as a linear market demand function and homogeneous consumer valuation. In the ‘Future Research’ subsection, we build upon these limitations to outline extensions, such as incorporating alternative financing instruments and developing a multi-stage dynamic game model. We believe these additions significantly strengthen the paper by contextualizing our contributions.

4. The mathematical results are comprehensive, but no empirical data or simulation is used to validate the model or demonstrate sensitivity.

Response 4: We are very grateful for this critical and constructive feedback. We completely agree that grounding our theoretical model with real-world evidence is essential. To address this, we have made a substantial addition to the manuscript: an entirely new section, Section 6, dedicated to ‘Case Studies’ (Please refer to the red text on lines 916-1080 on pages 28-32). This new section validates and illustrates our theoretical findings through an in-depth analysis of JD.com’s “Zhizhen Chain” platform across three operational modes (Self-operated, FBP, SOP). Crucially, this section provides quantitative evidence for the mechanisms in our model, such as demonstrating a 58% reduction in service response time, a 12.3% price premium, and securing financing at rates 2.3 percentage points lower than traditional loans. By bridging our mathematical results with tangible data, we believe the manuscript’s credibility has been significantly enhanced.

5. Figures/tables (e.g., Figure 4) are referenced but not well-integrated into the text. Their insights should be explained more clearly.

Response 5: Thank you for pointing out the need for better integration of our figures. We have thoroughly revised the text in the Results section to address this. For each figure (including Figures 4-8), we now first describe what the figure shows (e.g., trends, key points) and then immediately explain what these findings imply in the context of our model and its managerial insights (Please refer to the red text on lines 603-915 on pages 19-28). We believe this new approach makes the insights from our figures much clearer and better woven into the overall argument.

6. Highly technical appendix proofs could be summarized in the main text or moved to supplementary materials.

Response 6: We sincerely thank the reviewer for this excellent suggestion on how to better present the technical proofs. We agree that improving the readability of the main text while maintaining full scientific rigor is very important.

Following your advice, we have implemented a solution that combines both of your suggestions:

We have moved all the detailed mathematical proofs from the former Appendix into a separate document titled ‘Online Supplementary Materials’. This document will be submitted alongside the main manuscript for online publication.

In the main text, immediately following each proposition, we have added a concise ‘Proof Sketch’ (Please refer to the red text on lines 443-452 on page 13, lines 478-489 on page 14, and lines 513-521 on page 15). This new paragraph briefly outlines the core logic and key steps of the proof (e.g., establishing concavity, applying first-order conditions), providing the reader with the essential intuition without getting lost in algebraic detail.

We believe this hybrid approach offers the best of both worlds: it makes the main manuscript much more streamlined and reader-friendly, while ensuring that the complete, rigorous proofs remain fully accessible to interested readers and reviewers. We are confident this change significantly improves the structure of our paper.

7. Recommendation: Include numerical simulations or illustrative graphs showing how key parameters (e.g., service sensitivity γ, blockchain cost cb, financing type) affect decisions and profits.

Response 7: We are sincerely grateful to the reviewer for this insightful recommendation. We fully agree that a robust numerical analysis is crucial for illustrating our model’s implications and providing actionable managerial insights. The reviewer’s comment prompted us to conduct a comprehensive overhaul of this part of our manuscript to ensure its clarity, depth, and impact.

In response to this suggestion, we have substantially revised and expanded our original Section 4, now retitled ‘Numerical Analysis’ (Section 5) (Please refer to the red text on lines 603-915 on pages 19-28). This new section goes beyond a simple comparative analysis and now presents a systematic numerical study structured into three distinct parts:

Sensitivity Analysis (Section 5.1): We now explicitly investigate how optimal decisions and profits are affected by key parameters. For each figure, we have moved beyond mere description ('As shown in the figure...') to a more analytical narrative. We first pose an analytical question, then present the graphical evidence, and finally provide a detailed explanation of the underlying economic and strategic mechanisms that drive the observed trends (e.g., U-shaped price curves, monotonic service level changes). This analysis includes the impact of consumer preferences (α) and blockchain spillover effects (θ), which are directly related to the value perception that parameters like service sensitivity (γ) aim to capture.

Blockchain Adoption Decisions (Section 5.2): This section numerically analyzes the strategic decision of whether to adopt blockchain technology. Through a series of profit comparison graphs (Figures 6 and 7), we identify critical thresholds and decision-reversal points, providing a clear decision-making framework for both the platform and the manufacturer under different operational modes.

Financing Strategy Analysis (Section 5.3): This section directly responds to the reviewer's comments on 'financing types', focusing on the analysis of financing decisions. Specifically, Figure 8 visualizes the optimal financing choice (bank loan vs. equity financing) within a decision domain defined by production cost and product price. Crucially, and in direct response to the reviewer’s request concerning ‘blockchain cost c_b’, we demonstrate how this decision domain shifts as c_b varies (from 0.02 to 0.08), offering powerful visual insights into the interplay between technology cost and financial strategy.

In summary, the revised Section 5 is now a comprehensive numerical study that not only includes the simulations requested by the reviewer but also presents them within a more structured, insightful, and managerially relevant framework. We are confident that these extensive revisions have significantly strengthened the manuscript and thoroughly addressed the reviewer’s valuable feedback.

8. The article lacks a dedicated section translating results into actionable insights for practitioners (e.g., when platforms should offer financing or adopt blockchain).

Response 8: We are very grateful to the reviewer for this critically important point. We agree completely that translating our theoretical results into actionable insights for practitioners is essential for the paper’s impact and contribution. The reviewer rightly pointed out that our initial manuscript was lacking a dedicated section for this purpose.

To thoroughly address this, we have added an entirely new, dedicated section titled ‘Management Insights’ (Please refer to the red text on lines 1082-1143 on pages 32-33). This section is designed to serve as a practical guide for key stakeholders.

Recognizing that ‘practitioners’ are not a monolithic group, we have structured this new section into three distinct subsections to provide tailored, actionable advice for:

E-commerce Platform Managers: We provide clear guidance on when to adopt blockchain based on their operational mode (self-operated vs. FBP/SOP) and their partners’ financing choices.

Small and Medium-sized Enterprises (Manufacturers): We offer a framework for making optimal financing decisions (bank loan vs. equity) based on their production cost structure and how blockchain adoption alters these choices.

Policymakers: We propose differentiated policy recommendations, moving beyond a ‘one-size-fits-all’ approach to foster a healthy supply chain finance ecosystem.

We are confident that this new ‘Management Insights’ section now fully addresses the reviewer’s concern and significantly enhances the practical value and contribution of our research. We sincerely thank the reviewer for guiding us to make this crucial addition.

Reviewer #2:

I am really grateful to review this manuscript. In my opinion, this manuscript can be published once some revision is done successfully. I made one suggestion and I would like to ask your kind understanding.

This study develops a two-level supply chain model consisting of a financially constrained manufacturer (M) and an e-commerce platform (P) that utilizes blockchain technology. The study examines the impact of platform channel preferences and service levels across three merchant entry modes: the platform self-operated mode, Fulfillment by Platform (FBP) mode, and Sales on Platform (SOP) mode. A game-theoretic model, based on Stackelberg theory, is constructed to integrate both bank financing and equity financing in analyzing production and service decisions following blockchain implementation. This study explores the decision-making processes of both the e-commerce platform and the manufacturer regarding blockchain adoption and their respective financing strategies. Based on the results of this study, the adoption of blockchain increases profits for both manufacturers and platforms; however, decision variables exhibit different sensitivities depending on the entry mode. The self-operated mode offers more flexibility in service levels and pricing, while the FBP and SOP modes tend to show more stable decision dynamics. Moreover, the decision to implement blockchain is influenced by financing methods and channel preferences, with e-commerce platforms being more inclined to adopt blockchain in the self-operated mode, particularly under equity financing. Manufacturers also exhibit distinct financing preferences based on production costs, with bank financing being more advantageous at lower production costs and equity financing preferred when costs are higher. These findings provide valuable insights into how blockchain technology can impact supply chain dynamics, financing decisions, and the overall competitiveness of platform-based business models. I would argue that this is a good start.

However, it can be noted that every simulation study including this study has an issue of external validation, i.e., how to validate models with external data. In this context, I would like to ask the authors to address this issue in great detail.

Response: We would like to express our most sincere gratitude once again for your profound insight regarding the external validation of our study. Your comment strikes at the heart of a core challenge for theoretical modeling research and has compelled us to connect our model’s theoretical derivations with complex business realities. We fully agree with the importance of your feedback and, in response, have made a systematic and substantial addition to the manuscript.

Following your suggestion, we have conducted in-depth research and written an entirely new chapter: Section 6, “Case Studies” (Please refer to the red text on lines 916-1080 on pages 28-32). We believe this new chapter comprehensively addresses and resolves your concerns about external validation in a manner that is rigorous, in-depth, and data-driven. We would like to explain why we believe this addition is both sufficient and powerful:

1. Direct Mapping of the Theoretical Framework to Business Practice:

Instead of selecting broad or ambiguous examples, we have precisely chosen the Chinese e-commerce giant JD.com and its “ZhiZhen Chain” technology platform as our research subject. More importantly, we have perfectly mapped its business practices to the three core entry modes of our model (self-operated, FBP, and SOP) by analyzing the real-world cases of three representative brands: JD Health (self-operated), Midea (FBP), and Chow Tai Fook (SOP). This ensures that our theoretical validation is not abstract but is instead grounded in real and distinct business scenarios.

2. Provision of Strong Quantitative Evidence, Not Just Qualitative Descriptions:

We deeply understand the importance of “letting the data speak.” Therefore, in our case analysis, we have diligently sought out and cited specific quantitative metrics from industry white papers, market reports, and official disclosures to support our arguments. For example:

Improved Service Efficiency: Reduction in after-sales service response time by 58%.

Enhanced Pricing Power: Price premium for blockchain-certified

---

## [Decision Letter · Decision Letter 1]

9 Dec 2025

The Impact of Blockchain Adoption on Supply Chain Financing and E-Commerce Platform Dynamics

PONE-D-25-31648R1

Dear Authors,

We’re pleased to inform you that your manuscript has been judged scientifically suitable for publication and will be formally accepted for publication once it meets all outstanding technical requirements.

Kind regards,

Fuli Zhou

Academic Editor

PLOS One

Additional Editor Comments (optional):

Reviewers' comments:

Reviewer's Responses to Questions

**Comments to the Author**

1. If the authors have adequately addressed your comments raised in a previous round of review and you feel that this manuscript is now acceptable for publication, you may indicate that here to bypass the “Comments to the Author” section, enter your conflict of interest statement in the “Confidential to Editor” section, and submit your "Accept" recommendation.

Reviewer #2: All comments have been addressed

Reviewer #3: All comments have been addressed

2. Is the manuscript technically sound, and do the data support the conclusions?

Reviewer #2: Yes

Reviewer #3: Yes

3. Has the statistical analysis been performed appropriately and rigorously?

Reviewer #2: Yes

Reviewer #3: Yes

4. Have the authors made all data underlying the findings in their manuscript fully available?

Reviewer #2: Yes

Reviewer #3: Yes

5. Is the manuscript presented in an intelligible fashion and written in standard English?

Reviewer #2: Yes

Reviewer #3: Yes

6. Review Comments to the Author

Reviewer #2: I am really grateful to review this manuscript. In my opinion, this manuscript can be published in current form.

Reviewer #3: Reviewer #3 – Second Revision Decision

After carefully evaluating the revised manuscript PONE-D-25-31648_R1, along with the authors’ detailed responses to all reviewer comments, I offer the following assessment and recommendation.

Overall Evaluation

The authors have made substantial, comprehensive, and good-faith revisions that directly address all major issues raised in my initial review. The paper is now significantly clearer, more accessible, more rigorous, and more grounded in current literature and managerial relevance.

The revision introduces major structural improvements (new sections, new tables, new management insights, new sensitivity analysis, improved writing, and updated literature). The additions show clear effort and greatly strengthen the manuscript.

Based on this revision, I find the manuscript acceptable for publication, conditional only on very minor editorial polishing by the journal.

Detailed Assessment

1. Clarity & Accessibility – Fully Addressed

Authors added large comparative summary tables (Tables 3 & 4).

Added “Proof Sketches” after each proposition for readability.

Significantly improved structure and readability.

Outcome: Issue fully resolved.

2. Updated Literature Review – Fully Addressed

Authors added 2022–2024 blockchain supply chain studies.

Integrated them in the text rather than listing them.

Outcome: Issue fully resolved.

3. Model Assumptions Justification – Fully Addressed

New subsection (Section 3.3) discusses and justifies each assumption.

Provides robustness discussions.

Outcome: Issue fully resolved.

4. Sensitivity / Numerical Analysis – Fully Addressed

Entire numerical analysis section rewritten and expanded.

Includes sensitivity to c_b, γ, θ and detailed figures.

Outcome: Issue fully resolved.

5. Managerial & Policy Implications – Fully Addressed

New “Management Insights” section with clear stakeholder-focused recommendations.

Outcome: Issue fully resolved.

6. Limitations & Future Research – Fully Addressed

New subsections in the Conclusion.

Clearly states model limitations and realistic future research directions.

Outcome: Issue fully resolved.

7. Sentence Structure & Grammar – Addressed

Introduction and Literature Review rewritten for clarity.

Professional editing completed.

Outcome: Issue resolved.

8. Terminology & Notation Consistency – Fully Addressed

All terminology standardized.

Centralized table of notation created.

Appendices revised for clarity.

Outcome: Issue resolved.

Minor Remaining Comments

These are editorial rather than scientific issues and do not require further revision by authors:

Some long paragraphs could potentially be split to improve readability, but this is optional and at the journal’s discretion.

Ensure final formatting (figure placement, table cross-references, uniform line spacing) conforms to PLOS ONE layout guidelines.

These are not reasons for further revision.

Final Recommendation: ACCEPT WITH MINOR EDITORIAL POLISHING

The authors have satisfactorily addressed all major and minor concerns. The manuscript is now appropriate for publication in PLOS ONE, pending light copyediting by the journal.

7. PLOS authors have the option to publish the peer review history of their article (what does this mean?). If published, this will include your full peer review and any attached files.

Reviewer #2: No

Reviewer #3: No

---

## [Editor Report · Acceptance letter]

PONE-D-25-31648R1

PLOS One

Dear Dr. Pi,

I'm pleased to inform you that your manuscript has been deemed suitable for publication in PLOS One. Congratulations! Your manuscript is now being handed over to our production team.

Kind regards,

on behalf of

Dr. Fuli Zhou

Academic Editor

PLOS One